# Supervised Machine Learning and Deep Learning Techniques for Epileptic Seizure Recognition Using EEG Signals—A Systematic Literature Review

**DOI:** 10.3390/bioengineering9120781

**Published:** 2022-12-08

**Authors:** Mohamed Sami Nafea, Zool Hilmi Ismail

**Affiliations:** 1Computer Engineering Department, College of Engineering and Technology, Arab Academy for Science and Technology (AAST), Cairo 2033, Egypt; 2Malaysia-Japan International Institute of Technology, Universiti Teknologi Malaysia, Jalan Sultan Yahya Petra, Kuala Lumpur 54100, Malaysia

**Keywords:** EEG, machine learning, deep learning, epilepsy, seizure detection, systematic review

## Abstract

Electroencephalography (EEG) is a complicated, non-stationary signal that requires extensive preprocessing and feature extraction approaches to be accurately analyzed. In recent times, Deep learning (DL) has shown great promise in exploiting the characteristics of EEG signals as it can learn relevant features from raw data autonomously. Although studies involving DL have become more common in the last two years, the topic of whether DL truly delivers advantages over conventional Machine learning (ML) methodologies remains unsettled. This study aims to present a detailed overview of the main challenges in the field of seizure detection, prediction, and classification utilizing EEG data, and the approaches taken to solve them using ML and DL methods. A systematic review was conducted surveying peer-reviewed publications published between 2017 and 16 July 2022 using two scientific databases (Web of Science and Scopus) totaling 6822 references after discarding duplicate publications. Whereas 2262 articles were screened based on the title, abstract, and keywords, only 214 were eligible for full-text assessment. A total of 91 papers have been included in this survey after meeting the eligible inclusion and exclusion criteria. The most significant findings from the review are summarized, and several important concepts involving ML and DL for seizure detection, prediction, and classification are discussed in further depth. This review aims to learn more about the different approaches for identifying different types and stages of epileptic seizures, which may then be employed to enhance the lives of epileptic patients in the future, as well as aid experts in the field.

## 1. Introduction

Epilepsy is a brain disease defined primarily by frequent and unpredictable disruptions in normal brain activity, causing what is known as epileptic seizures. An epileptic seizure is a brief period of abnormally elevated or concurrent neuronal activity in the brain. Some epileptic seizures may involve unprovoked jerking of the body or loss of awareness, both of which can result in a brief loss of control that can lead to serious injury or even death while performing dangerous activities. According to the World Health Organization (WHO), epilepsy affects over 50 million individuals worldwide, who are of varying ages and ethnic backgrounds [1]. For around 70% of this population, anti-epilepsy medications can keep their conditions under control [2]. On the other hand, around 30% are poorly responsive to such treatments, requiring surgical intervention [3]. As reported, there is a significant shortage of neurologists and the neurological services needed by these personnel, which can greatly affect the timely delivery of treatment to patients [4]. Therefore, automatic recognition of seizures is vital in order to aid neurologists and allied healthcare providers expedite the process of patient diagnosis, and prescribe the required treatments, if applicable.

Electroencephalography (EEG) was first introduced by Hans Berger to measure the electrical activity of different regions in the human brain, which can be particularly useful in the diagnosis of different types of brain disorders. Such a tool helps neurologists study the fluctuations in the brain that occur during epileptic seizures. The analysis of these fluctuations can aid in accurately distinguishing between healthy and unhealthy functionalities of the brain. To properly analyze epileptic seizures, long-term EEG data spanning days, weeks, and occasionally months is required, which demands a significant amount of human time and effort. In the literature, different epileptic seizure recognition tasks, using machine learning (ML) and deep learning (DL) approaches, are categorized as follows:Seizure detection is where a model identifies the presence or lack of seizures or abnormal activities after analyzing EEG signals.Seizure prediction refers to the ability of a model to predict the likelihood of the occurrences of imminent epileptic seizures early on, by identifying the patient’s preictal state.Seizure/Phase classification is where a model is able to categorize different types of seizures or seizure phases. In other scenarios, the *classification* term is used for classifying different seizure phases, known in the literature as EEG/phase classification.

Figure 1 depicts an abstract overview of detection, prediction, and classification tasks of epileptic seizure recognition tasks. Seizure detection can be performed when it is required to review EEG recordings and evaluate seizure occurrences. The presence of a seizure is detected in Figure 1a. Seizure prediction is often required to alert health care professionals or patients of upcoming seizures, so safety precautions can be taken before the seizure’s commencement. As shown in Figure 1b, an alarm is raised before seizure onset. Seizure classification can assist neurologists determine the type of the seizure, which can be crucial for taking the appropriate medical decision [2]. As shown in Figure 1c, the seizure is identified and its type classified.

In order to comprehend the relationship between these tasks and epileptic seizure analysis, it is necessary to recognize the various phases of an epileptic seizure using EEG as shown in Figure 2. The interictal phase describes the baseline state of the brain between two consecutive seizures, when no seizure activity is present [5]. The preictal phase is theoretically the period before the brain enters into a seizure; however, it is not considered a part of a seizure. According to Frank et al. [6], the preictal phase is usually accompanied by an inexplicable sensation that can last several hours or even days before the seizure. Around 20% of the patients experience this phase symptomatically, which may help them take precautions prior to seizure onset. The following phase is the ictal phase that most epileptic patients encounter. This phase is characterized by intense electrical activity in the brain and is usually symptomatic. Most of the ictal phases last between 30 s and 2 min. The postictal phase is considered the recovery period after a seizure ceases before returning to the baseline.

Considering the significance of recognizing epileptic seizures and their various phases, the notion of automating the procedure began to develop. The first paper using a computer algorithm to detect seizures was published in 1991 [7], while the first paper employing artificial neural networks was in 1996 [8]. ML has made substantial progress in tackling research problems during the 1990s, coinciding with the rise of computers. Artificial neural networks algorithm (ANN) is a DL approach, a subfield of ML, consisting of a sequence of algorithms that replicate the way the human brain works.

Due to increased computer processing power and storage, ML and DL algorithms can process large amounts of data, enabling the discovery and extraction of usable knowledge. Figure 3 demonstrates the rise in the use of EEG recordings in the study of epileptic seizures. Therefore, this paper aims to present a systematic literature review (SLR) covering the challenges, solutions, and employed methods for epileptic seizure detection, prediction, and classification using EEG signals.

The rest of the paper is structured as follows: Section 2 presents a background on the main steps for both machine learning and deep learning architectures. Section 3 describes the research methodology adopted for planning and executing this SLR. Section 4 presents the results acquired from the survey and highlights the journals where most of the articles are published. Section 5 discusses the challenges, solutions, and employed methods. In Section 6, public datasets that can be used for epileptic seizure research are described in detail. Finally, Section 7 concludes this review by highlighting the contributions and summarizing the concluding remarks.

## 2. Background on Machine Learning (ML) and Deep Learning (DL)

ML is the process of training a computer to use its previous exposure to data to solve problems that are presented to it. Because of the current availability of lower-cost processing power and memory, the concept of applying ML in several domains to solve problems faster than humans has attracted many researchers. Due to the availability of such resources, it is now possible to process and analyze extremely vast amounts of data to reveal insights from and correlations between the data that are difficult to see with the naked eye. ML cognitive behavior is built on algorithms that allow the machine to form abstractions based on prior knowledge. For ML to work effectively, it relies on manually created, handcrafted, features to be extracted from the data, which necessitates experts in the domain of the problem. DL has a more advanced approach that allows automatic data extraction relying on multi-layer structures. Generally, the performance of ML or DL is measured through a classification algorithm. Even though ML has a variety of classification techniques, and the results are relatively good, DL is taking over.

In the field of epileptic seizure applications, building a model involves multiple steps, which are EEG data acquisition, data preprocessing, development of a machine learning or deep learning model, and a final performance evaluation step. During the EEG data acquisition step, electrodes are placed on the human head to capture EEG signals through special equipment. This data is composed of different readings for each electrode, usually called a recording channel, and is stored for relevant use. The data preprocessing step involves data cleaning such as removing artifacts, removing noise from the signal, omitting missing records, and data normalization.

The feature engineering step in the ML approach is divided into two stages: feature extraction and feature selection. During the feature extraction step, handcrafted features are created from the raw data. These features shall be meaningful for the ML model; thus, it is important that they are discriminative and non-redundant so the data can be thoroughly exploited. In the feature selection step, all these features are combined, and the best features are selected while also reducing the number of features (dimensionality reduction). The final step is the classification step, where a classifier is used to categorize the data.

In contrast to ML’s approach, DL uses a deep neural network to perform the entire process. It makes use of several nonlinear activation units distributed across multiple layers. When the processing of a unit is finished, its output is fed into the next one. Moving through the hierarchical data structure, each level processes its data into a more abstract form that may be fed to the next level, which performs the automatic feature extraction and feature selection steps. The final layer is typically used as a classifier, with the activated unit representing the decision to be made. Figure 4 presents the classical model of the automated epileptic seizure recognition framework for both ML and DL.

A related topic to ML and DL is transfer learning (TL), which aims to transfer the knowledge obtained from learning one problem to another problem that is different but still related, thus avoiding the learning process all over again. This is done by re-using already pre-trained models for the new task, which greatly speeds up the training process. TL is split into 3 categories, which are inductive TL, transductive TL, and unsupervised TL. Selecting one of these categories solely depends on whether the data labels are available or not for both the source and target domain. The reader can refer to [9] for more details about transfer learning techniques.

## 3. Research Methodology

A comprehensive and systematic literature review approach has been frequently employed to find, assess, and interpret significant research on certain subjects, research scopes, or phenomena. Reviewing studies with the same scope, this strategy seeks to assess the challenges addressed and solutions implemented. This review was conducted systematically while conforming to preferred reporting items for systematic reviews and meta-analysis (PRISMA) guidelines [10]. PRISMA is an evidence-based minimum set of items for reporting in systematic reviews and meta-analyses. A planning protocol has been formed and consists of 4 stages: (1) research questions, (2) execution procedure, (3) screening for eligible studies, and (4) data extraction process.

### 3.1. Research Questions

It is necessary to understand the research questions and their significance before carrying out the execution procedure. The following are the research questions regarding the employment of ML and DL techniques in the field of epileptic seizures using EEG:RQ1: Is the recognition task involved in detection, prediction, or classification of epileptic seizures?RQ2: What are the ML/DL techniques applied to achieve any of these tasks?RQ3: What is the data used to achieve any of these tasks?RQ4: What are the challenges present during the application of ML and DL techniques to achieve these tasks?RQ5: How is ML/DL going to impact the clinical practice of epileptic seizure analysis?

### 3.2. Execution Procedure

The execution procedure was carried out on 16 July 2022. It was performed using a string of terms that are commonly used in the literature related to the required tasks in the field (e.g., detection, prediction, classification, EEG, epileptic, seizure). For this SLR, a specific combination of keywords is used to formulate the key term-based search for the survey using Web of Science and Scopus databases, which is “EEG” AND “Epileptic” AND “Seizure” AND (“Detection” OR “Classification” OR “Prediction”). Review articles, books, editorials, and conference papers were excluded, in addition to any articles written in any other language than English. Figure 5 shows the number of publications included for full-text assessment (dark bar), and the remaining eligible publications to be reviewed (light bar) for both the Web of Sciences and Scopus databases.

### 3.3. Eligible Studies

In order to gather relevant data for this systematic literature review, all relevant papers that addressed the research questions were examined. The number of eligible papers for the study is extracted according to the following criteria:Inclusion criteria

I1: Published in an indexed journal.

I2: Published between 2017 and the date of the execution procedure.

I3: Focuses on epileptic seizures through the application of EEG signals.

I4: The work is conducted using EEG on human brain signals.

I5: The type of EEG signal acquisition is either intracranial (iEEG) or scalp (sEEG).

I6: The full text of the paper is accessible.

I7: The work is empirical (not a survey or a review).

Exclusion criteria

E1: Implemented techniques other than ML/DL.

E2: Used non-public datasets.

E3: Employed a non-supervised ML/DL learning technique.

E4: Involves analysis of other signal recording types besides EEG (MEG, MRI, etc.).

E5: Implementation offers no new state-of-art performance, no new methodology, or just an application of ML/DL over EEG data.

Data extraction fields

In order to synthesize and draw conclusions from the studies evaluated for eligibility, relevant data shall be extracted and analyzed. The following data fields are extracted during the process:

D1: The challenge that the paper aims to solve.

D2: Employed ML or DL technique, either classical or a combination of techniques.

D3: The feature engineering technique(s) used for signal processing.

D4: The epileptic seizure recognition task: whether detection, prediction, or classification.

D5: The dataset utilized for the performed task.

## 4. Results

In this section, the findings of the systematic literature review are presented. A total of 8165 articles were collected according to the execution procedure. After removing 1343 duplicate articles, a total of 4560 articles remained to be initially screened. However, since this SLR solely focused on the preceding 5 years, starting from year 2017 till the date when the survey was conducted on 16 July 2022, the articles left for the first screening stage were 2262. The first screening stage was performed by screening the titles, abstracts, and keywords of the articles using Endnote software. During the screening process, if the article matched any of the inclusion criteria (I1 to I7), it was included for full-text retrieval. 2043 articles did not meet all the inclusion criteria; thus, they were excluded.

In the second screening stage, 219 articles remained for assessment; however, 5 articles could not be retrieved and were thus omitted, leaving 214 articles eligible for full-text assessment. The remaining 214 articles were thoroughly inspected to ensure their compliance with the eligibility criteria before deciding whether to include them for review or not. A total of 123 publications were excluded since they met one or more of the exclusion criteria (E1 to E5).

At the end of this stage, 91 articles were eligible for review, and the required data according to the Data Extraction criteria (D1 to D5) was extracted from the full-text for analysis. Figure 6 overviews the whole process of article searching and selection according to the PRISMA protocol.

### 4.1. Article Distribution among Journals and Publishers

The eligible articles chosen for the study are based on both supervised ML and DL techniques applied to recognize epileptic seizures using EEG data published across 46 different journals, including engineering and bioinformatics specializations. Among these journals, *Biomedical Signal Processing* is the top journal with 12 out of 91 publications, followed by 5 publications for each of *IEEE Access*, *Biocybernetics and Biomedical Engineering, Computers in Biology and Medicine*, *Journal of Biomedical and Health Informatics,* and *Transactions on Neural Systems and Rehabilitation Engineering*. The published articles belong to a broad range of journals including biomedical engineering, bioinformatics, neurosciences, signal processing, and computer sciences, which focus on artificial intelligence applications. In addition to *IEEE Access*, which is a multidisciplinary journal, some journals focus on different engineering aspects such as *Chaos, Solitons and Fractals*, *Sensors*, and *IEEE Sensors.* Publishers-wise, *Elsevier* has the greatest number of articles, with 36 publications across 13 distinct journals, totaling 40% of the articles included in this review. *IEEE* follows with 26 publications across 11 different journals, and *Springer* comes next with 10 publications across 7 different journals. Figure 7 depicts all the included journals and their corresponding publishers.

### 4.2. Articles Distribution among Epileptic Seizure Recognition Tasks

After the analysis of all the eligible papers using the previous screening criteria in Section 3.3, the number of articles for each of the detection, prediction, and classification tasks is previewed in Figure 8. A summary of the recent work includes the number of papers performing these tasks between 2017 and 16 July 2022. The analysis reveals that the most recent work focuses on the seizure detection task, with a total number of 38 publications. The seizure prediction task received attention as well, particularly in 2021 with 10 publications, while both seizure detection and classification tasks are gaining traction in 2022. The classification task statistics involve both seizure and seizure phase classifications. Based on the executed survey, Table 1 lists the top 10 cited articles performing the detection task on epileptic seizure data.

### 4.3. Article Distribution among Employed Machine Learning and Deep Learning Techniques

Figure 9 shows the number of eligible articles that employed either ML or DL techniques for epileptic seizure detection, prediction, and classification tasks. This survey shows that ML techniques were employed in many more publications than DL techniques before 2021. One possible explanation for this is that ML algorithms are easier to build and require less powerful hardware than DL techniques. Despite this, DL techniques are gaining prevalence due to their high performance. Table 2 presents the top 5 cited articles for both ML and DL techniques.

## 5. Discussion

Several studies have utilized EEG data either for the detection, prediction, or classification of epileptic seizures. Many researchers have tried to interpret EEG recordings for clinical and scientific purposes, with varying degrees of success. Throughout this section, a summary of the challenges, solutions, and methods employed will be discussed.

### 5.1. Discussion of the Challenges

#### 5.1.1. EEG Signal Complexity and Data Transformation

EEG signals exhibit chaotic and non-linear dynamics, where subtle variations in the signal cannot be detected through human-eye inspection. Additionally, scalp EEG recordings (sEEG) are notorious for noises and artifacts due to slight motor movements such as eye blinks, cardiac signals, and muscle movements [23]. To deal with these noises and artifacts, preprocessing techniques are usually required to clean the signal.

Because of the way machine learning approaches function, providing EEG signals directly impacts the ability to properly abstract meaningful descriptions from the signal; consequently, reliable handcrafted features must be extracted from the signal. These features should be well correlated with different seizure phases to achieve optimal performance. Current feature extraction methods involve transforming the signal into different domains using different methods. These methods range from simple statistical analysis to complex non-linear methods [24]. On the other hand, approaches that depend on deep learning techniques do not require manually created features.

#### 5.1.2. High Number of EEG Channels and Channel Optimization

EEG signals are multidimensional time-series data with various numbers of EEG channels. Channels refer to the electrodes located on the scalp of the head or inside the skull. Despite 64 channels being common in research, the number of EEG channels can vary from as few as 2 channels up to 256 channels. The process of setting up many electrodes is a tremendous and time-consuming task. Additionally, as the number of channels increases, the computing power needed to store and process the data also increases, since more data samples are collected over time.

Overfitting is another issue that may develop as a result of the use of excessive redundant channels. A study has shown that a maximum number of 35 channels can be enough for a full EEG montage [25], while other publications found that the number of channels can even go down to 3 channels for epileptic seizure recognition while mitigating such a phenomenon [26].

Moreover, due to physiological variances among subjects, there is considerable inter-subject variability [27,28], rendering the reduced selected channels sub-optimal for most individuals. Such an issue may hinder the performance of seizure recognition when tested among various subjects. Therefore, selecting the appropriate channels that extrapolate across the majority of the subjects is a challenging task.

#### 5.1.3. Generalization Ability

Variability in signal patterns, physiological differences, and the scarcity of seizure events in EEG data pose a difficulty for automated epileptic seizure models to operate efficiently across different patients. Developing a patient-specific model requires prior patient knowledge which can considerably improve overall seizure diagnosis performance, especially if medical treatment is needed. Nevertheless, the performance of patient-specific models when new unseen patient data is introduced is adversely impacted. This performance loss is mainly due to the overfitting of the patient-specific model to the seen patient data, leading to weak decision boundaries [29,30]. Such an approach obliges clinicians to record new EEG data specific to new patients.

While generalization across different patients is favored, the tradeoff between accuracy and generalization has always been an issue [31,32]. Additionally, the process of developing a model for each patient is not scalable as the number of patients grows. Although model generalization across large patient cohorts is a complex task, clinicians consider it more practical, despite a marginal performance hit [32].

#### 5.1.4. Data Imbalances

During an EEG recording session, seizure samples are significantly infrequent compared to non-seizure ones. Since seizure samples represent the minority class, highly imbalanced data poses a challenge because most ML or DL models will demonstrate a bias towards the majority class, and in severe circumstances may entirely overlook the minority class. Several methods to mitigate the adverse effects of data imbalances will be discussed.

### 5.2. Discussion of the Solutions

This section aims to present the solutions offered by eligible articles in this survey to provide insights into the methods and approaches used to resolve the previously mentioned challenges. It is vital to note that several methods may be frequently utilized while yielding different results. These results depend on the authors’ techniques and parameters, which are not always reported, affecting the reproducibility of the results. Table 3 summarizes the proposed solutions to the reviewed challenges in this survey.

#### 5.2.1. Signal Engineering

The transformation of the EEG signal is imperative to provide the necessary knowledge and to improve the analysis feasibility for ML and DL models. By extracting various features, the analysis of EEG signals can be performed using data from different signal domains, such as time domain, frequency domain, and time-frequency domain features. These features can also be categorized as linear and non-linear features [33]. Entropy is another type of feature that can be extracted from various signal domains to measure the irregularity and unpredictability of signal fluctuations. Additionally, complex techniques such as wavelet analysis are often used to analyze EEG signals [34].

##### Time Domain Features

Statistical properties, such as mean, median, variance, standard deviation, skewness, kurtosis, peak amplitude, minimum amplitude, peak to peak, and similar, are the simplest features that may be derived from an EEG signal in the time domain [22,35,36,37,38,39,40,41,42,43]. Hjorth parameters are based on the variance of the subsequent derivatives of the EEG signal. The most used parameters are the first three derivatives of the signal, which are activity, complexity, and mobility, measuring the variance, the changes in the frequency, and the mean frequency, respectively [35,41,44,45,46]. Such statistical properties enable the epileptic seizure recognition model to characterize the signal.

Other time domain features are energy, coefficient of variation, and peak-to-peak amplitude. Energy represents the sum of square magnitudes of the amplitude of the signal; coefficient of variation represents the ratio of dispersion; and peak amplitude measures the deviation between the maximum and minimum peaks of a signal [35]. These features may be used as an indicator for the variation in the activity of the EEG signals. Some of these statistical features can also be extracted from MinMax histograms of the EEG signal through quantifying spikes and sharp fluctuations to identify seizure events from histogram bins [44]. In [15], common spatial pattern (CSP) paired with convolutional neural networks (CNN) were proposed to predict seizures by distinguishing preictal from interictal segments. CSP was used to extract features from several frequency sub-bands, which were decomposed using wavelet packet decomposition. The purpose of CSP is to maximize the variance of one class while minimizing the other’s, thus increasing the borderline between these classes. Afterward, the output is projected to form a covariance matrix. In [47], an algorithm based on filter bank CSP was proposed to extract discriminative patterns from raw signals. It was combined with 2D-CNN and layer-wise relevance propagation (LRP) to extract features from the frequency sub-bands, and enable the interpretation of the predictions through relevance scores, respectively.

##### Frequency Domain Features

Fast Fourier transform (FFT) is an efficient variant of discrete Fourier transform (DFT), which is used to transform a raw EEG signal from the time domain to the frequency domain. In [48], FFT was used to convert raw EEG signals to generate a matrix consisting of the absolute values of frequency amplitudes. In [49], the raw multivariate signals were transformed into spectrograms, where FFT is considered an essential computation part to generate spectrograms. In [12,40], FFT was applied on windowed time segments from each EEG channel to extract frequency information. In [20], FFT was used to handle narrow band signals due to its speed and reliability. In [50], two architectural styles were used. The first employs a variant of FFT named short-time Fourier transform (STFT) to convert the raw signal into the time-frequency domain, while the second architecture utilizes both FFT and principal component analysis (PCA) to convert the raw signal into separated time and frequency domains. In [51], FFT was computed over the five main frequency ranges of the EEG signal, ranging from delta to gamma bands. The aim of the study was to compare the robustness of frequency domain features against time-frequency domain features.

Power spectral density (PSD) is a measure of the distribution of power of the discrete frequencies that compose the signal retrieved by applying Welch’s FFT method. In [13,14,20,35,52], PSD was used to analyze the frequency power in various brain states, which have discriminative ability between seizures and non-seizures. In [53], PSD and additional statistical features were applied after the signal was transformed into a time-frequency image (TFI). Non-seizure activities had low PSD with a non-stationary pattern, but seizure activities had high PSD values at certain frequencies with spike-wave patterns, demonstrating the potential of PSD in detecting seizures. Other power spectral measures include absolute and relative spectral powers, which were used to investigate the phase shifts between interictal and preictal states of the brain [52,54].

##### Time-Frequency Domain Features

Time-frequency features include a variety of techniques for capturing various aspects of EEG signal correlations. While some are theoretically linked, others are fundamentally different yet equivalent in terms of the information they convey regarding brain activity. Although time-frequency domain features provide more information compared to the separated time and frequency features, the resolution of the information presented may be compromised [55].

STFT overcomes some of the limitations of FFT as it calculates the frequency components over uniform short window intervals (widowing), preserving the time domain information for each window. In [56], STFT was used to transform the raw EEG signals into the time-frequency domain, and then a 2D TFI was constructed to describe the EEG signal. Local binary pattern (LBP) was applied over the images, revealing exceptional properties. Multiple domain features were retrieved from the raw signal, STFT, FFT, and discrete wavelet transformation (DWT) domains to provide multi-domain input data to produce a fuller representation [57]. In [58,59], deep neural networks consisting of a CNN and a long short-term memory recurrent neural network (LSTM-RNN) were employed to extract deep features and classify different types of seizures with remarkable F1-scores. In [60], spectrograms were fed to a gated three-tower transformer network (GTN), where the analysis is done time-wise, frequency-wise, and channel-wise. In [61], different pairs of window sizes and overlap percentages were tested to find the optimal setting for maximizing information with minimal redundancy.

Spectrograms are created from STFT transformations, constructing TFIs that visualize the frequency spectrums of a signal. This method can be beneficial when CNNs are used as feature extractors, so 2D TFIs are fed to the network instead of 1D time-series data [49]. A synchroextracting transform, an algorithm based on the Chirplet transform (CT) that uses STFT [62], was introduced to extract both instantaneous amplitude and frequency information. It was argued that the signal characterization in the frequency domain becomes fuzzy due to the uniform interval of the time window required. The proposed method claims to address STFT’s shortcomings in that area by improving the performance of epileptic seizure classification.

Since Fourier transforms lack the ability to capture localized frequency variations, the capacity to discern transient features over short periods is limited. Wavelet transforms (WT) solve this problem by simultaneously analyzing the signal’s time and frequency domains, allowing the extraction of local and transient components without compromising resolution [57]. DWT is one of the most widely applied wavelet transforms in the field of EEG signal analysis. In [20,35,63], DWT was used to obtain most of the brain’s rhythmic frequencies. Wavelet coefficients, named detail and approximation coefficients were extracted to capture the low and high frequencies in the wavelet with a varying number of filter banks.

Fine-tuning of DWT has also been performed by means of a weighted sliding window method based on the rate of energy measured. In [37], this method has been applied to the EEG signal while computing DWT to enhance the signal and suppress noisy components, thus improving the accuracy of the trained model. In [64], soft thresholds were computed based on each wavelet’s components, enhancing seizure zone localization. Redundancy removed dual-tree DWT (RR-DTDWT) [65] was proposed to overcome the information redundancy and the drawbacks of DWT. Dual-tree DWT (DTDWT) mitigates shift sensitivity, poor directionality, and the loss of phase information that DWT lacks, minimizing information loss. However, these benefits come at the expense of increasing information redundancy, which may degrade the performance of the epileptic seizure model. Hence, RR-DTDWT was developed to remove globally redundant information.

Wavelet packet decomposition (WPD) or wavelet packet transform (WPT) is a DWT extension that can provide more detailed frequency resolution by creating wavelets with full detail and approximation coefficients at each level. Conversely, DWT provides the same number of detail coefficients at each level in addition to one approximate coefficient. This difference enables detailed analysis of smaller frequencies. Additionally, it is not computationally expensive compared to continuous wavelet transform or Stockwell transform [16]. In [66], WPD was utilized to decompose the signal and reduce spatial redundancy and losses, followed by an epilepsy locality preserving projection algorithm (E-LPP) to reduce the dimensions of the extracted features. In [19], WPD was used to extract time-frequency features as a part of the multi-view system. Throughout, frequency and time-frequency features were combined into one framework to improve the generalizability of the seizure detection model across different patients. In [67], WPT was applied to fractionally Fourier transformed signals (FrFT), which expands the conventional Fourier transform with a rotation angle that describes a blend of both the time and frequency domains of the signal. Since FrFT and WPT were coupled, discriminative coefficients were obtained, resulting in better performance results for recognizing epileptic seizures. In [68], WPD is used to form vectors that are computed from each sub-band energy value across each channel. The purpose of this approach is to build an information reconstruction space to capture the subtle changes that occur during the preictal state, aiming to investigate the impact of various features on the encoding of the graph network.

Frequency-slice wavelet transform (FSWT) was utilized to decompose the signal into multiple sub-bands without requiring the design of any kind of bandpass filter banks, which facilitates the extraction of arbitrary frequency information [69]. Since FSWT has more adaptable time-frequency aggregation, the time-consuming procedure of creating custom filter banks can be avoided. Tunable Q-factor wavelet transform (TQWT) is an upgraded variant of DWT that allows customizable filter sizes, enabling transformations to be tuned based on the signal oscillatory characteristics. TQWT was used to decompose the raw signal into several sub-bands, followed by extracting several fuzzy entropy features [70] and Hjorth parameters [45]. This combination has enhanced the performance of seizure classification. Unlike DWT variants, continuous wavelet transform (CWT) is radically different as it deconstructs the signal across the entire time axis. Scalograms are absolute values of CWT which are sensitive to noise and fluctuations in the EEG signal. TFIs can be constructed from CWT coefficients, similar to STFT, providing a method which leverages more seizure-related characteristics, thus improving seizure prediction accuracy [71,72,73]. While CWT provides higher spectral resolution than DWT variants, it is more computationally expensive and produces redundant coefficients that increase the number of features. Additionally, CWT tackles the STFT’s fixed window size that hinders its ability to track signal dynamics. In [53], the decomposed coefficients were constructed in terms of mean-standard deviation wavelet coefficients (MS-WTC) to reduce the dimension of the generated features.

Another type of wavelet transforms, namely empirical wavelet transform (EWT), has been employed. The main difference between EWT and DWT is that EWT decomposes the signal using adaptive frequency boundaries for each wavelet based on the information content, in contrast to DWT which uses fixed frequency bands [74]. In [75,76], experiments involving DWT and EWT have shown that the adaptivity of EWT slightly improves seizure detection since the DWT’s fixed filter band efficiency is limited. Although EWT showed a significant improvement, its performance was still restrained by noisy and highly non-stationary signals [77]. Hence, a Fourier-Bessel series expansion EWT (FBSE-EWT) was proposed to overcome the drawback of standalone EWT. FBSE avoids the effect of widowing for frequency representations, which reduces distortions of the analyzed signal in the time domain. FBSE frequency representations require a coefficient number equal to the length of the analyzed signal, whereas FFT requires only half of the signal length, which results in greater frequency resolution, allowing minor fluctuations to be detected. In [11], EWT was extended to multivariate signals, called 2D-EWT, which utilizes cross-channel interdependence to recognize EEG seizures. The 2D-EWT decomposes signals into adaptive frequency sub-bands and utilizes the Hilbert transform (HT) to extract instantaneous features such as amplitudes, frequencies, and phases from each channel.

Stockwell transform (ST), commonly known as S-Transform, intertwines the capabilities of both CWT and STFT. CWT is not scale-invariant, so phase information is distorted, resulting in only locally referenced information [78]. ST excels at retaining the original signal’s phase information in the spectral domain, allowing complex signals to be properly characterized [79]. Furthermore, ST is less computationally expensive than CWT and provides significantly higher decomposition resolution than both STFT and DWT. Despite the advantages of ST, it afflicts the analysis when consecutive frequency components are overlapping, and chirp-like signals occur during seizures [80]. Generalized Stockwell transform (GST), accompanied by singular value decomposition (SVD), was presented to effectively overcome the limitations of ST in distinguishing ictal periods [18]. SVD was utilized as it is robust to noise, allowing useful information to be retrieved without using any noise removal techniques.

##### Non-Linear Features

The dynamics of the brain are both nonlinear and complex. Nonlinear components that cannot be broken down without affecting dynamics are considered complex. One of the most widely used non-linear features for analyzing EEG signals is fractal systems. Fractal systems consist of self-similar structures exhibiting behaviors from nature. Since the brain is a conceptually fractal entity, fractal dimension (FD) analysis is useful in analyzing the complexity of brain signals. Previous studies show that the self-similarity of brain waves varies depending on the brain state. During ictal events, signals are more self-similar and the degree of the fractal dimension differs across normal and epileptic patients depending on the signal duration [81,82,83]. Different fractal dimension algorithms that are commonly employed are Higuchi fractal dimension (HFD), Petrosian fractal dimension (PFD), and detrended fluctuation analysis (DFA). PFD is used to quantify the signal’s fluctuation and self-similarity while also displaying superior temporal resolution over FFT [44]. HFD is a particularly effective method for computing fractal dimensions since it is computationally fast while providing low calculation errors [84]. In [44,51], DFA was utilized as it can statistically quantify the self-affinity of non-stationary time-series signals through the values of the Hurst exponent. The Hurst exponent, also called the self-similarity exponent, reveals information about the signal fluctuation trend [85]. Line length (LL) is considered a simpler variant of FD [86], which has shown promising results in identifying burst changes in the signal that are commonly associated with seizures [35,76].

Empirical mode decomposition (EMD) decomposes non-stationary signals into a sum of component functions called the intrinsic mode function (IMF), which extracts oscillatory features that focus on subtle changes that occur in the signal [87]. In [88], both EMD and DWT were employed to extract features from their decompositions. Instantaneous energy, Teager energy, HFD, and PFD were extracted from each decomposed IMF and wavelet. Alternatively, in [89], several statistical features were extracted. Combining both EMD and DWT features yielded more information about the signal, resulting in better seizure classification. In [90], EMD was utilized as it had been claimed to perform better at denoising EEG signals. Several statistical features were extracted after rejecting decomposed IMFs that were below a certain signal-to-noise ratio (SNR) threshold. Spectral analysis showed that noisy high-frequency components of the signal had been removed while still retaining important seizure information. Other features like ellipse area of second-order difference plot (SODP) and fluctuation index were extracted from the signal IMFs [91]. It was observed that ellipse areas related to ictal IMFs are more prominent than the seizure-free IMFs due to the higher variation and fluctuation in the ictal IMFs. This feature has demonstrated the capability to distinguish between ictal and interictal events in phase space [92]. In [21], a wavelet-based envelope analysis (EA) method was proposed. This method employs DWT instead of EMD for signal decomposition, followed by applying envelope analysis using HT to the decomposed wavelets. Statistical features were computed over both the raw signal and the envelopes. It was claimed that the classification accuracy of the DWT-based EA method achieved slightly higher performance than EMD-HT.

Non-linear mode decomposition (NMD) is more robust to noise than EMD as it decomposes the signal adaptively by combining several time-frequency techniques. NMD yields only the meaningful physical oscillations of the signal without information loss in the form of non-linear mode (NM) sets. In [93], fractional central momentum (FCM), paired with NMD, was utilized to extract features from the NMD domain. In addition to its simplicity and fast computation speed, FCM demonstrated its capability to broaden the classification-related information between ictal and non-ictal events. Wavelet scattering transform (WST) algorithm is conceptually similar to CNN, but with fixed filter coefficients. WST layers wavelet transform, nonlinear modulus, and averaging operators to process the signal while providing translation-invariant time and frequency resolutions. In [94], the authors exploited the similar-to-CNN functionality of WST to decompose the signal into multiple sub-bands, from which several entropy features were extracted. The key advantage of WST is that clinicians can interpret the extracted features.

Recurrence plots (RP) are a graphical approach that visualizes the non-linear nature of a time-series signal through phase space. RP exploits the states of a dynamic system by constructing a two-dimensional binary matrix that corresponds to multiple points in the phase space trajectory that are roughly in the same region [95], thereby discovering hidden recurring patterns in the provided signal. In [96], the EEG signal was transformed to RP before being fed to an ensemble architecture of CNNs paired with a voting classifier. It was claimed that the use of RP displayed high-performance results in exploiting the interclass variability. The brain-rhythmic recurrence map (BRRM) [97] and unthresholded RP with fractal weighted LBP (URP-FWLBP) methods [98] were presented as improved versions to offset the loss of dynamical information due to the binarization process of RP. In [97], EEG signals are decomposed into three sub-bands and then transformed into images. The images are fed to a CNN-based architecture for classification. In [98], FD and LBP are combined to construct the images, followed by histogram analysis to extract feature values resembling the signals. Linear discernment analysis (LDA) was used for dimensionality reduction, and SVM was used for classification. The proposed method showed superior performance in seven evaluation metrics.

Spatial covariance matrix is a similar technique that employs matrices that are symmetric and positive definite (SPD), hence it belongs to the non-Euclidian Riemannian manifold. These matrices are transferred to the Euclidean domain as vectors using Riemannian geometry. In [99], SPD was utilized and shown to be less prone to noise and outliers while being able to identify seizures with high accuracy. In [100], the authors combined the capabilities of Riemannian geometry and fractals using the Riemann–Liouville fractional derivative (RLFD) operator to exploit the EEG signal in the continuous-time domain, resulting in special models for healthy and epileptic patients. Ictal events were characterized by histograms with heavy-tailed distributions.

The Mel-frequency cepstral coefficients (MFCC) technique is intensively used in speech recognition and seismological applications, where cepstral coefficients are generated in the log-spectrum domain. In [101], it was demonstrated that MFCC can be a valuable biomarker for identifying preictal events at an early stage due to the high variation that occurs in the feature maps. This representation is particularly important for seizure prediction tasks. In [102], MFCCs were extracted and fed as features to a Generalized Regression Neural Network (GRNN) with very excellent seizure classification performance.

##### Entropy Features

Entropy, which originated from information theory, is widely used to quantify the amount of disorder and chaos in a system using distribution probabilities, and hence, it can be used to measure the randomness of patterns in EEG signals. Some entropy features can be directly calculated in the time domain, whereas others require the signal to be transformed to the frequency domain or the wavelet domain.

Shannon’s entropy (ShEn) measures the uncertainty and randomness of time-series data in correspondence to the logarithm of the number of possibilities. In [29], ShEn was used to compute the uncertainty of data samples from an annotated pooling dataset. If the entropy value is above a certain threshold, samples are not confidently identified by the current iteration of the proposed algorithm, and further training is required. It has also been applied to signal histogram data calculated via PSD to quantify the complexity of the signal [13] and was used as one of the features in [52]. Spectral entropy (SEN), commonly called power spectrum entropy, is calculated on the normalized PSD values using classical Shannon’s entropy to quantify the spectral complexity of the signal in the frequency domain [13,35,39,52]. In [48], it was observed that healthy patients had lower median spectral entropy than epileptic patients.

Wavelet entropy (WE) is slightly similar to SEN; however, the wavelet decomposition coefficients are used instead to compute the relative energies across EEG signal frequency bands. WE evaluates stimuli responses in different frequency bands, with wider bands resulting in high entropy values [103]. In [20,44,52], WE was used to quantify the behavior of the EEG signals. Log Energy entropy (LogEn) is similar to Wavelet entropy, except that it only employs the sum of logarithmic probabilities. In [52], LogEn was applied on raw signals, while in [76] and [94], it was applied on the sub-bands of the DWT coefficients and the sub-bands generated from the scattering wavelet transform, respectively. In both the latter articles, LogEn contributed significantly to seizure detection.

Renyi’s entropy (REN) is a generalization of Shannon’s entropy. Mathematically, SEN is considered a special case of Renyi’s entropy, where it differs from SEN in the lower frequency bands while remaining similar in the higher frequency bands. In [62], REN was employed to calculate the energy concentration in the proposed methods, where a lower REN value indicates a higher presence of energy concentration. In [39], along with other time-frequency features, REN was used to differentiate between seizure and non-seizure events. However, not much information was conveyed when REN was used in standalone. In [56], a KECA, a variant of KPCA [46,104], was introduced, in which the principal components are chosen based on their contribution degree to REN entropy.

Approximate entropy (ApEn) computes the irregularities in a signal without requiring prior knowledge about the source of the data, making its applications nearly unlimited. Sample entropy (SampEn) is considered an improvement over approximate entropy [105]. Furthermore, lower SampEn values imply that the signal is self-similar, whereas larger values indicate higher complexity. It is claimed that signal complexity declines during seizure activity, which means the measure of sample entropy can be a potential for seizure detection [106]. In [39], ApEn was used to calculate the randomness of the signal, and it was found that healthy patients had the highest median value and epileptic patients with ictal activity had the lowest. In [35], ApEn was calculated for samples with less than a certain threshold, while SampEn was calculated for other subsequent samples higher than that threshold. Both ApEn and SampEn are capable of contributing effectively to the detection of seizure activities [44,51,52,64,65,66]. Multiscale entropy (MSE) is an extension of sample entropy and is used to compute signal complexity when multiple time-series scales are involved, particularly when the signal time-series relevance is unknown. Before the entropy computation, a coarse-graining process is performed, which allows multiple temporal scales to be investigated. Modified multiscale entropy (MMSE) is similar to MSE; however, the coarse-graining process is calculated using the moving-average procedure, providing better complexity analysis. In [65], MMSE was utilized as an indicator for seizures.

Distribution entropy (DistEn) was proposed to mitigate the shortcomings and parameter dependency in both ApEn and SampEn when applied to small datasets. It measures the time-series complexity by applying an empirical probability density function (ePDF) of inter-vector distances in the data state space. In [107], DistEn showed a potential capacity to distinguish between ictal and interictal events. Fuzzy entropy (FuzzyEn) is comparable to ShEn and SampEn, although FuzzyEn measures irregular signal uncertainties while ShEn measures probabilistic uncertainties [108]. In [107], FuzzyEn performed well with varying EEG data segment lengths in discriminating between healthy and epileptic individuals, but not ictal and interictal events. In [70], 15 fuzzy entropy-based features were extracted to construct a feature set, and the seizures were classified using an adaptive neuro-fuzzy inference system (ANFIS). Combining fuzzy features with ANFIS resulted in better classification performance than using non-fuzzy features. In [70], FuzzyEn, combined with LogEn, was applied to the sub-bands generated by the WST domain. Due to the exponential function continuity, FuzzyEn was able to effectively reflect the intrinsic patterns included within the ictal EEG signals. In [16], a combination of FuzzyEn and DistEn called Fuzzy Distribution entropy (fDistEn) was developed. The hybrid entropy technique integrates the ePDF and fuzzy membership similarity degree to eliminate the strict boundaries caused by the ePDF. fDistEn was able to exploit the complexity of EEG signals and produced statistically significant results when compared to standalone DistEn and FuzzyEn.

Singular value decomposition entropy (SVDEn) decomposes a signal into a sum of independent components, which enables locating regularities in temporal and spatial domains. Unlike spectral entropy, the processing is done on the singular spectrum, which is robust to noisy signals [18], in addition to having the ability to locate the patterns of ictal signals [35,51]. Permutation entropy (PE) is a measure of the non-stationarity of a signal based on calculating the repetition of occurrences of neighboring values. The computation of PE is fast and simple; thus, it requires less data preprocessing, and it can be directly applied to large datasets. Furthermore, it is robust to noises and was therefore employed as one of the features that contributed to seizure detection [44,51,52].

##### Cross-Correlation Features

Because EEG signals are typically acquired from different parts of the brain where channels are placed, investigating the multivariate dynamics across brain regions can be extremely valuable. Investigating cross-correlation features utilizing the interaction between multi-channels can assist in understanding the underlying brain dynamics compared to univariate analysis of individual channels.

Phase-locking value (PLV) is a metric used to measure the synchronicity between two time-series, quantifying their phase interactions. The value of PLV approaches zero in the case where the two signals are independent of each other and have a uniform distribution. Contrarily, if the phase of the two signals is strongly coupled, the value of PLV approaches one [109]. PLV was used to derive the strength of the spatial correlations between EEG channels and to construct the graph edges for the proposed Graph Neural Network (GNN) method [110]. In [29], GATENet, a sparse self-gating mechanism, was utilized to capture abnormal activities in epileptic patients. When both randomized and PLV methods were used to generate graph edge features, PLV performed better than randomly constructed graph edges; nonetheless, GATENet performed better than both.

Phase-lag index (PLI) was proposed to mitigate the shortcoming of PLV regarding its sensitivity when a common source is shared between channels. This is done by eliminating the phase differences that are centered around zero. Weighted phase-lag index (wPLI) is an extension of PLI and considers the magnitude of the lag in calculating the phase difference. Such merit helps in decreasing the possibility of false positives in the case of existing noise near zero-phase locking, as well as in increasing the detection rate of phase synchronization. In [111], PLI and wPLI were utilized to predict impending seizures. The trend of both metrics significantly rose during the beginning of the preictal phase, with wPLI reaching its peak during seizure onset. It should be noted that PLV, PLI, and wPLI are often calculated using the EEG signal time-derivatives rather than raw signals to facilitate the discrimination between the interictal and preictal phases [112].

The Pearson correlation coefficient (PCC) measures the linear similarity between two random variables, quantifying the magnitude of the correlation between −1 and 1. If the values assigned are close to the boundaries, it means a strong positive or negative correlation exists, while a value of zero means no correlation. Likewise, mutual information (MI) quantifies the uncertainty of mutual interdependency between two random variables and is calculated using relative Shannon entropy or the Kullback–Leibler (KL) divergence. Both methods are widely used in the literature during the feature selection stage to select highly correlated features to enhance the performance of detecting seizures. In [41,90], PCC was used as a feature selector to minimize the size of the feature set, since a high number of features may severely impact the model’s learning performance. Alternatively, PCC has also been used to compute the correlation between different EEG channels, creating a matrix to assemble a brain network graph. Similarly, MI was used to select the best feature set before the classification process [91]. In [11,62,113], MI was used to compute the similarity between all EEG channels in order to find the optimal seizure detection channels. In [73,101,114], KL was used to measure the divergence between the distribution of features extracted from both interictal and preictal samples to identify where the phase shift had occurred.

Coherence measures the synchronization of the spectral components’ activity between observed channels. In [115], coherence was applied to the eigenvalues distribution matrix for each time window, followed by calculating the spectral-based covariance matrix. Low values imply good channel coherence, whereas high values suggest random states.

#### 5.2.2. EEG Channel Reduction and Attention

Generally, multiple channels are usually required to investigate the whole dynamics of the brain in multiple regions. However, the analysis of data from all channels is usually not required. Depending on the research application, signal processing techniques may be required to extract meaningful features from these channels, resulting in high computational load and data explosion due to high EEG temporal resolution. The main purpose of reducing the number of channels is to reduce the computational complexity of the performed task, so less data is consumed. Furthermore, by reducing the number of channels, relevant channels with the most significant features are selected, omitting redundant data thus ﻿mitigating overfitting issues. Additionally, the procedure may be required in some applications such as wearable devices where using a large number of channels is impractical [26]. Channel selection can be performed using different approaches, whether they are statistical approaches [11,22,36,62,75,76,113], data-driven approaches [14,88,116,117,118], wrapper approaches [119], or from prior knowledge based on previous studies [120,121].

*Statistical approaches*: Several articles [11,62,76,113] followed the same principle to select the five best channels for optimal seizure detection. Since seizure events appear for a very short duration in long recordings, seizure events may be recognized by their spike patterns. Although artifacts can resemble the same spike pattern, they usually persist for longer durations. As a result, if the rate and amplitude of these spikes are repeatedly protracted, they can be analytically distinguished from seizure events by computing the standard deviation (SD) of the signal. Because a high SD may indicate that the spikes are artifacts, the channel with the lowest standard deviation is chosen as the seed channel. Following that, MI between the seed channel and the remaining channels is evaluated, and the channels with the highest similarity to the seed channel are selected. According to [75], calculating the kurtosis of all the channels and selecting the highest values is a better alternative since the former method implies that the channel with the lowest SD has the highest SNR. Nevertheless, this can also indicate that this channel contains very few seizure events. Additionally, channels selected in correspondence to the MI of the seed channel may contain redundant data. Although employing kurtosis resulted in higher performance than using SD and MI, the difference is statistically insignificant. In [22], an iterative process is run across the entire dataset, computing the product of the variance and entropy for all channels. The detection model is trained and tested on an initial set of channels sorted in descending order based on the variance–entropy product. For some patients, channels with the same or better accuracy are chosen, bringing the average number of channels down to 10. Similarly, in [36], the variance was calculated for all the channels, and the three channels with the highest variance were selected.

*Data-Driven approaches*: Random Forest (RF) algorithm was utilized to identify the most informative channels [14], by integrating all the extracted features from all the channels into random RF-generated trees. Because significant features are expected to emerge more often than redundant ones, the channels corresponding to those frequent features were selected. It was demonstrated that employing this technique obtained a marginal separation between seizure and non-seizure data after the t-SNE algorithm was applied, achieving a close F-measure score using the entire set of channels against only three channels. Another method based on optimization techniques such as genetic algorithms (GA) was utilized to reduce the number of channels to a single channel while minimally affecting the detection performance [88]. The process involves extracting the features from each channel for all the patients, followed by the non-dominated sorting genetic algorithm (NSGA), which creates the populations of chromosomes for every channel. The ones and zeroes in the chromosome signify which channels along with their corresponding features will be evaluated. The highest accuracy reported across four different classifiers is chosen. The method is iterated until the termination requirements are met. The study evaluated two GA variants, NSGA-II and NSGA-III, with the latter outperforming the former.

Other data-driven approaches involve using attention mechanisms that can automatically attend to informative channels by assigning attention scores to these channels during the model’s training process. In [117], maximum and average pooling operators are applied to the input feature matrix, followed by a convolution layer. The outputs are vertically concatenated, and a non-linear sigmoid activation function is then applied to form the channel attention score. This procedure results in weighted feature maps corresponding to different channels, where features with high weights are considered more important to the seizure prediction task. A similar concept was introduced in [63,116], but a non-linear softmax activation function was used instead of a sigmoid one. Likewise, in [118,122], the mechanism attends to different channels by learning the attention weights through incorporating multiple layers of convolution layers followed by fully connected layers.

*Wrapper approaches*: In [119], the method of backward elimination was utilized. The method involves computing the accuracy of the model while leaving out a single channel at a time, ruling out channels that have no detrimental impact on prediction performance. The procedure is repeated until removing any more channels impacts the prediction accuracy negatively.

#### 5.2.3. Adjustable Training Approaches, Data Augmentation, and Various Learning and Training Techniques

Throughout the survey, various strategies were implemented to improve the ability of models to generalize across data. This includes, but is not limited to, different regularization techniques [117], early stopping [58,123], data augmentation [50], ensemble learning [90], transfer learning [22], multi-view learning [19,57,124] and subject-independent approaches [44,101,125,126]. Since discussing the approaches to regularization is a broad topic, it is considered beyond the scope of this review. However, the reader may refer to [127,128] for more information.

In [117], the batch normalized LSTM (BN-LSTM) technique was used. The difference between LSTM and BN-LSTM is that the latter handles covariance shifts that happen between the hidden-to-hidden layers, boosting the model’s generalization and convergence [129]. In [58,123], the training process was stopped if no apparent improvement was observed after a pre-determined number of epochs. The early stopping method is typically effective in preventing the trained model from over-fitting to the training data. It should be noted that determining the criteria for early stopping is not an easy task, because it is rather trivial to stop the training process earlier than required.

Adversarial training, a domain adaptation technique based on adversaries, aims to inject adversarial samples into the training data so the model begins to learn from these samples. In [49], transferable features (adversaries) were created by a classifier–discriminator in a two-player min–max game. The discriminator must distinguish between training and test data samples while the classifier (generator) is trained to extract new transferable features and manipulate the discriminator. This process regularizes the model to resist overfitting and increases its generalization ability across unseen data. Similar concepts were adopted using squeeze-and-excitation networks (SENet) [31] to extract spatiotemporal features from two different datasets with different domain distributions. The model was able to generalize well in cases when the training data was sufficient. However, the performance drastically dropped in the case of domain variation and insufficient training data. In [118], both the seizure and patient data are separately decomposed and reconstructed forming two modules that compute a reconstruction loss value. The training process measures the loss in the signal reconstruction of both modules, aiming to enhance the generalization across different patients.

Data Augmentation is another approach that artificially augments the input samples by injecting distortions and random noises during model training. This approach has two advantages: (i) it can be used to efficiently increase the number of samples during the training phase; and (ii) it forces the model to learn the semantics of the data, instead of just memorizing patterns. Therefore, data augmentation can be utilized to improve generalization capabilities. Generative adversarial networks (GAN) are a machine learning technique that comprises two neural networks, named the generator and the discriminator. Both networks compete in an adversarial game, where one network (the discriminator) aims to distinguish real data from generated ones, while the other network (the generator) keeps generating new enhanced data. GAN is a prominent technique capable of generating new synthetic data through learning the real data’s statistical characteristics, and it has gained notoriety with time-series data. Due to the lack of data to efficiently train DL models, GAN was employed to generate new samples for preictal EEG signals [130]. The approach consists of the generator creating new samples while the discriminator continuously inspects whether the newly created samples are real or fake. The accuracy performance was significantly improved by including the synthetically generated signals in the dataset. A different approach is to induce perturbations that cause minor changes to the EEG signal without affecting its semantics. In [50], the spectrograms were subjected to random masking and Gaussian augmentation. A subset of the spectrogram content in each sample was randomly chosen and discarded, and then Gaussian noise was generated and placed in the discarded position.

Ensemble learning combines the predictions of different models to form the final prediction output. Ensemble learning boosts the overall performance since different models are unlikely to make the same errors when trained on the same dataset. Furthermore, ensemble learning can be boosted via bagging techniques. This method introduces different subsets of the training dataset to the different models in the ensemble, thereby reducing the high variance and improving the stability of the training process. In [90], SVM, CNN, and LSTM were used as ensemble classifiers incorporated with a model-agnostic meta-learning (MAML) technique. MAML is a supervised technique for few-shot learning approaches that can learn new tasks from small amounts of training data with a minimal number of gradient updates. Ensemble classifiers, along with MAML, were combined to predict epileptic seizures with great generalization ability and a low false positive rate.

Multi-view learning (MVL) is a paradigm that seeks to learn common modalities and patterns by combining several features from distinct domains to obtain discriminative representations of the data. In [17], a group convolution SE block (gcSE) was proposed to combine the multi-domain features extracted from different sub-bands to extract heterogeneous features. The framework combining gcSE and SENet enhanced the detection performance. In [19], multi-view features are extracted from raw signals and signals that are transformed by FFT and WPD. Afterward, the deep multi-view features are generated using CNN before being fed to a multi-view Takagi, Sugeno, and Kang fuzzy system (MV-TSK-FS) classifier. It is claimed that feeding the classifier with data from different feature spaces increases the model’s generalizability. In [57], a multi-view paradigm with few-shot learning is adopted. CNN was used to extract deep features from raw signals and signals transformed by DFT, STFT, and DWT. This module is followed by a feature-fusion mechanism to concatenate all the features forming a new feature representation. It was evident that leveraging the multi-domain features resulted in better detection performance.

Transfer learning allows using the knowledge of previously trained models, which can vastly boost the generalization ability when new models are fine-tuned. In [22], using a pre-trained deep convolutional auto-encoder (DCAE) provided a significant reduction in the number of trainable parameters, reducing the training time while improving generalization. Similarly, InceptionV3, a CNN architecture that is widely used in image recognition and object detection, was used as a transfer learning model for classifying different seizure types after transforming EEG signals to spectrograms, with relatively good results [131].

Moreover, structural modifications have been introduced to CNN to allow the architecture to become more robust. In [38], variable tuning blocks were integrated between adjacent convolutional layers instead of depending on static weights, on the basis that static weights limit the learning scope of the network, thus limiting its ability to generalize. To address this issue, two different designs with dynamic weighing mechanisms and special weight-tuning blocks were introduced. The proposed designs aimed to change the variable weights depending on the nature of the input data. Both models significantly improved the classification performance; nevertheless, it was explicitly stated that the dynamic weighing mechanism caused a massive overload due to the increased number of trainable parameters, affecting the training speed. In [71], semi-dilated convolutions were introduced to leverage the rectangular-shaped geometry of EEG scalograms, where the larger dimension is exploited rather than both dimensions. The proposed architecture has been shown to improve the generalization and robustness of seizure prediction. In [97], the basic CNN cell has been modified to include point-wise and depth-wise convolutions with a residual connection to reduce required parameters. The proposed modification improved the generalization ability compared to the basic CNN cell.

In contrast to subject-specific approaches, subject-independent approaches involve designing and training the models to capture the seizure patterns irrespective of the data distribution. In [44], different feature selection algorithms were combined to select the most contributory features from all the patients for seizure detection. The subset of features was optimized, and redundant features were omitted. In [101], two multi-task architectures, CNN and a Siamese network, were proposed. Both networks involve learning the seizures and patient-related information. The Siamese network was able to separate patients well, discerning their differences when being trained as a preictal phase classifier, which improved seizure prediction. In [125], a graph synthesizing network in combination with GNN and LSTM was used to generate brain graphs and learn feature embeddings for seizure prediction. The generated graphs improved the system’s ability to learn seizure patterns irrespective of the patient. In [126], the model is trained using a combination of multi-scale convolution and a spatial-temporal feature extraction module. The model was able to generalize as it learned features from different convolutional scales. The leave-one-out cross-validation (LOOCV) strategy has been adopted to boost the generalizability of the model on unseen patient data [115,126].

#### 5.2.4. Data Resampling and Class Balancing

During EEG recording sessions, seizures and seizure-free episodes will usually be the minority and majority of events, respectively. Therefore, balancing the dataset among different classes is essential to mitigate the bias towards the majority class during the training process of ML or DL models. Strategies including resampling (data-level) balance the number of samples for the minority class, while others such as ensemble/weight-tuning (algorithm-level) assign weights to the minority class samples to reduce the bias.

Oversampling is a concept that aims to generate new samples derived from the same distribution as the minority class until the dataset is balanced. Oversampling can be as simple as randomly replicating samples from the minority class [115] or as complex as synthetically generating new samples derived from the same data distribution. Undersampling is a strategy to randomly reduce the number of majority class samples until both class samples quantitatively match [73].

Splitting signals into segments is quite a common practice in preparing EEG data. The segmentation process can be done with or without overlapping partitions of each subsequent segment in order to create more segments. The process is repeated until the desired number of segments is obtained (oversampling). A *t-seconds* window is defined to determine the overlapping duration of each segment. This approach has been employed in [37,41,60,62,117,121,132] to balance the desired class samples. Depending on the length of the segments, the number of created samples via overlapping may be insufficient to balance out the dataset, hence under-sampling can be used as an additional step to balance the class samples [117]. Conversely, in [116], manual class distribution was done since the dataset contained eight different classes; thus, it was necessary to maintain an adequate class distribution across different cross-validation folds. In [13,17,38,44,53,91,119], the quantity of seizure and non-seizure samples were equalized for the model’s training phase (undersampling).

Another approach to creating new data samples is synthetic resampling. Synthetic resampling is used to generate new synthetic samples that comply with the distribution of the resampled class. Synthetic minority oversampling technique (SMOTE) [133] is a technique that generates new samples by determining the k-nearest neighbors of every minority class sample, then randomly creating new linearly correlated samples between the designated sample and its neighbors. In [11,75,76], SMOTE was used to generate new samples and balance the dataset. An adaptive synthetic algorithm (ADASYN) [128] is similar to SMOTE, with a primary difference being that it takes into account the local distribution of the class to be oversampled and scatters the new samples by adding some variation to them. ADASYN was employed in [48,93] to augment the number of minority class samples.

Since GAN can generate new synthetic data from the original dataset, it can also be employed to balance the number of samples among different classes. Conditional GAN (CGAN) is a type of GAN that leverages the available labels in the dataset to create new similar structures to the provided sample. In [90,134], GAN was employed to create synthetic seizure samples to balance the training dataset. Similarly, creating new adversarial features solved the data imbalance problem in the training dataset [49].

Some other approaches do not depend on oversampling or undersampling. These approaches give more weight to the minority class in order to reduce the bias toward the majority class [52,115,126]. Similarly, *focal loss* is an objective (loss) function that allows the training model to alleviate the impact of data imbalance by focusing on the minority class while reducing the weight of the majority class. Using *focal loss* has resulted in higher seizure detection [135] and prediction [61] performance. Another approach has included signal segmentation and recombination in different domains [15,136].

Several variants of boosting algorithms have been purposefully developed to deal with the imbalanced data issue [137]. AdaBoost focuses on dealing with the misclassified instances rather than the imbalance of the data itself. In [138], Adaboost was combined with least-square SVM to address the issue of unbalanced data by boosting the weights of misclassified samples while decreasing the weights of correctly classified ones. Consequently, the overall classifier was boosted through multiple weak classifiers. In [121], the cascaded architecture utilizing Adaboost with different classifiers was able to deal with the class imbalance and improved the performance of seizure detection.

Finally, several performance evaluation metrics such as F-measure, ROC curve, Matthew’s correlation coefficient (MCC), G-mean, and Cohen’s kappa are preferred for evaluating the performance of models on imbalanced data, since both precision and recall metrics are taken into consideration [58,123].

### 5.3. Performance Comparison

In this sub-section, the results of 10 recent articles, published in 2022, using ML and DL techniques are presented. Table 4 provides a summary of the selected articles and their best achieved performance results across different datasets.

## 6. Public Datasets for Epileptic Seizure Tasks

To evaluate any of the seizure epileptic detection, classification, or prediction models, it is essential to have an EEG dataset containing diverse seizure recording sessions. Table 5 provides a detailed list of frequently used public EEG datasets used in seizure recognition tasks.

### 6.1. CHB-MIT

CHB-MIT is an sEEG multichannel dataset obtained from Children’s Hospital Boston Massachusetts Institute of Technology (CHB-MIT), and publicly accessible through PhysioNet [139]. The dataset comprises 977 h of scalp EEG (sEEG) recordings utilizing 23 bipolar channels (some recordings are 24 and 26) placed according to the International 10–20 electrode positioning system with a sampling frequency of 256 Hz. The recordings were collected from 23 pediatric patients: 5 males whose ages are between 3 and 22 years, and 17 females whose ages are between 1.5 and 19 years, in addition to one anonymous patient. Each patient has between 9 and 42 EEG recordings stored in EDF file format, where each recording lasts for a duration of 1 h. However, some recordings are up to 4 h long.

### 6.2. TUSZ

The TUH EEG Seizure dataset (TUSZ) is considered the largest open-source dataset so far that focuses on epileptic patients and is a subset of the Temple Hospital University (TUH) EEG dataset [140]. The dataset features high-quality annotations for eight different epileptic seizure types, along with the patient’s detailed metadata describing the patient’s medications and clinical history. The dataset includes 1400+ h of EEG recordings using 24 to 36 channels (19 channels are common) stored in EDF file format. For the EEG recording sessions, a bipolar temporal central parasagittal (TCP) montage with two common reference points, average reference (AR) and linked ear (LE), is used. The dataset is occasionally updated with new data (the most recent version is v1.5.4) and is freely available upon acquiring login credentials from the corpora owners.

### 6.3. Bonn

The Bonn dataset is collected under the supervision of the University of Bonn [141] and consists of five sets of EEG recordings, where the first two sets (A and B) are captured from healthy subjects, and the other three sets (C, D, and E) are captured from five brain surgery candidates. Sets A and B vary in the state of the healthy subjects during the recording session with their eyes open (set A) and closed (set B). Sets C and D are EEG recordings in the interictal state from two different brain regions: the hippocampal (set C) and an epileptogenic zone (set D), whereas set E contains only ictal state recordings. Each set consists of 100 single-channel EEG recordings with a duration of 23.6 s each, stored in textual file format. All the segments are preprocessed using a band-pass filter with a 0.53 Hz to 40 Hz cut-off frequency. The initial recording configuration, as per [141], used 128 channels; nevertheless, relevant data about the patients and channels was not included.

### 6.4. Bern–Barcelona

This dataset was obtained from Pompeu Fabra University, Barcelona [142], and it consists of pairs of EEG signals that were captured from five patients who had undergone surgical resection. The dataset contains 3750 pairs comprising 7500 segments at a sampling rate of 512 HZ with a 20 s segment each, stored in textual file format. Each pair is composed of focal and non-focal segments that form two time-series signals captured from two adjacent intracranial channels from epileptogenic and non-epileptogenic zones, respectively. A fourth-order Butterworth band-pass filter between 0.5 Hz and 150 Hz has been applied to all the EEG recordings to reduce phase distortions.

### 6.5. NSC-ND

This dataset is a subset extracted from a private dataset that belongs to the Neurology and Sleep Centre (NSC), Hauz Khas, New Delhi [143]. The EEG recordings have been recorded from 10 epileptic patients using gold-plated scalp electrodes positioned according to the international 10–20 placement system. The recordings are sampled at 200 Hz and preprocessed using a band-pass filter with cut-off frequencies between 0.5 Hz and 70 Hz. The dataset consists of three sets of 50 single-channel EEG recordings, 5.12 s each, stored in MATLAB file format. The sets are categorized into preictal, interictal, and ictal stages.

### 6.6. SWEC-ETHZ

SWEC-ETH dataset was recorded during a pre-surgical analysis of epileptic patients at the Sleep-Wake Epilepsy Center (SWEC) of the Department of Neurology at the University of Bern and the Integrated Systems Laboratory of the ETH Zurich [144]. The intracranial scheme was set up using strip, grid, and depth electrodes. All the iEEG signals are pre-processed using a fourth-order Butterworth band-pass filter between 0.5 Hz and 150 Hz cut-off frequencies, sampled at 512 Hz, and stored in MATLAB file format. The dataset comprises 2656 h of recordings for 18 patients using 24 to 118 recording channels. Each EEG recording is divided into three sections that have been carefully analyzed by professionals. The recordings consist of 3 min of preictal activity, followed by an ictal activity that ranges between 10 and 1002 s. The ictal activity is followed by another 3 min of postictal activity.

## 7. Conclusions

In this systematic literature review, different articles have been explored, covering different approaches for automatic EEG seizure detection, classification, and prediction using ML and DL techniques following the PRSIMA protocol. Four challenges have been assembled, each of which presents the main issues encountered during seizure analysis tasks. Several works addressing these challenges and introducing the approaches to solving them have been thoroughly discussed. It should be noted that signal transformation is a challenging process because it is heavily dependent on the nature of the data and the artifacts contained in the signals. Since no method works for all types of EEG data, combining different approaches, as reviewed, can greatly aid in mitigating the shortcomings of some methods, resulting in higher recognition performance. Additionally, channel selection is becoming a crucial task to decrease the computational burden as well as to create wearable seizure detection applications. However, this significantly affects the process of seizure localization since the physical electrodes corresponding to these channels are no longer existent.

Deep Learning began to emerge as an indispensable tool in the field of neurology and EEG seizure analysis due to its ability to exploit EEG data more thoroughly and extract features without preprocessing while achieving high recognition performance. However, DL models are well-known for their black-box nature, which conceals their inner workings. As a result, neurologists and clinicians are becoming more skeptical about the interpretability of these models, which has an adverse effect on their use in clinical settings [145,146]. As a result, research including approaches to interpreting the predictions of DL algorithms, such as explainable artificial intelligence (XAI), is required to boost confidence in their use. Furthermore, despite the promising results of both ML and DL techniques, generalization among unseen patients from different datasets is still problematic due to the data domain difference. This problem leads to a high variation in seizure recognition performance, which contributes to experts’ skepticism regarding the performance of these techniques in real-world scenarios [145,146].

Finally, several employed epileptic seizure datasets are reviewed. TUSZ is the only dataset, so far, that includes a wide range of seizures and extended recording hours, allowing ML and DL techniques to be adequately trained. Conversely, most studies use small datasets that may not be large enough to accurately reflect the performance of their proposed work in real-world scenarios. Moreover, despite the TUSZ dataset comprising several types of epileptic seizures, some seizure terms do not conform to the most recent nomenclature as determined by the International League Against Epilepsy (ILAE). Simple partial and complex partial seizures are examples of these seizures, which are currently designated as focal aware and focal impaired awareness seizures, respectively [147]. Moreover, these types of seizures require further clinical evaluation as they cannot be solely identified through EEG analysis. These limitations emphasize the importance of increased collaboration between bioengineers and neurologists, as well as discovering new approaches, using ML and DL, that incorporate the use of clinical reports along with EEG analysis to gain deeper knowledge about similar types of seizures.

## Figures and Tables

**Figure 1 bioengineering-09-00781-f001:**
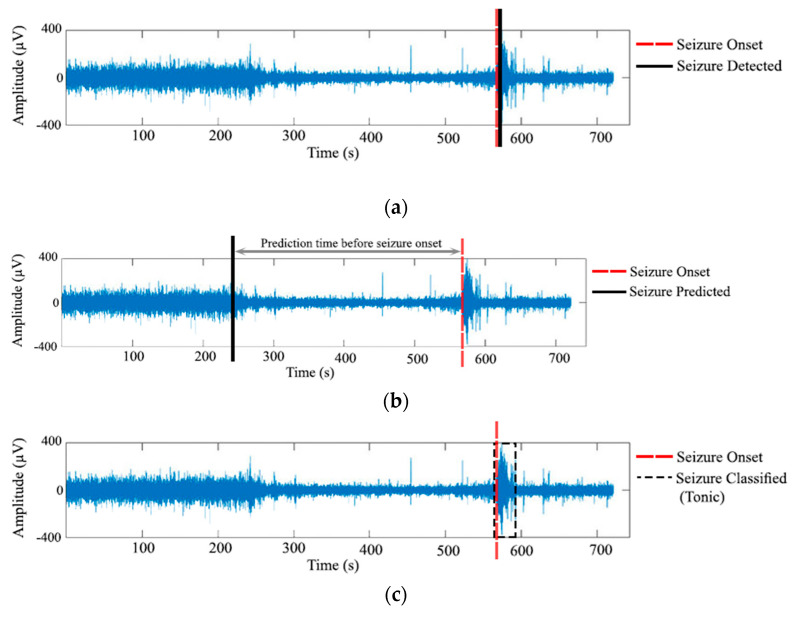
Different recognition tasks for diagnosis of epilepsy: (**a**) seizure detection; (**b**) seizure prediction; (**c**) seizure type classification.

**Figure 2 bioengineering-09-00781-f002:**
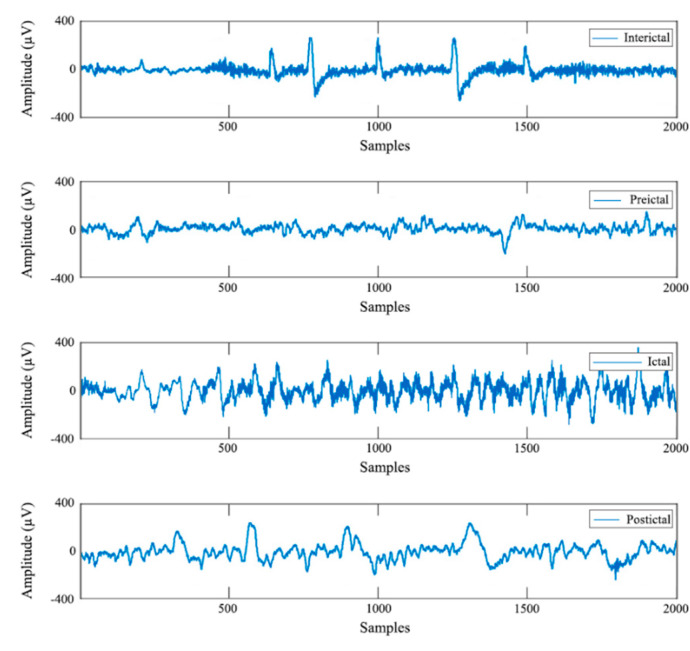
The different phases of epileptic seizures.

**Figure 3 bioengineering-09-00781-f003:**
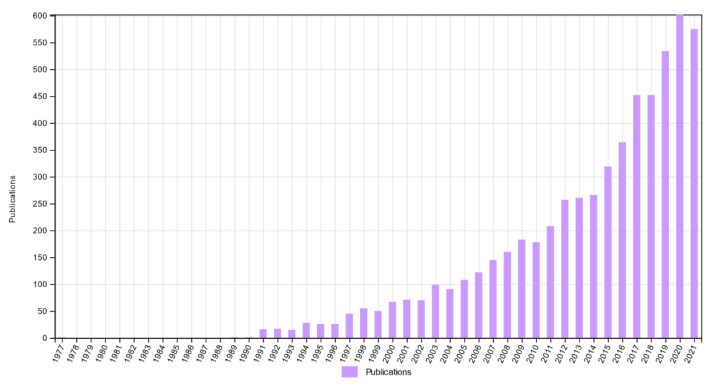
Number of articles on seizure detection, prediction, and classification published since the 1970s till 2021 as reported by Web of Science.

**Figure 4 bioengineering-09-00781-f004:**
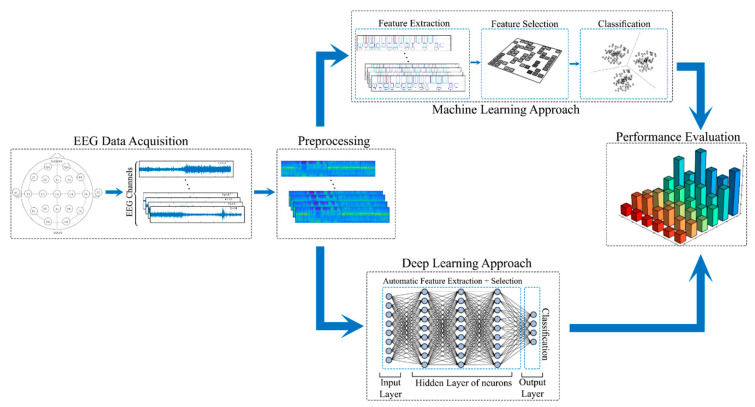
Diagram for classical automated epileptic seizure model.

**Figure 5 bioengineering-09-00781-f005:**
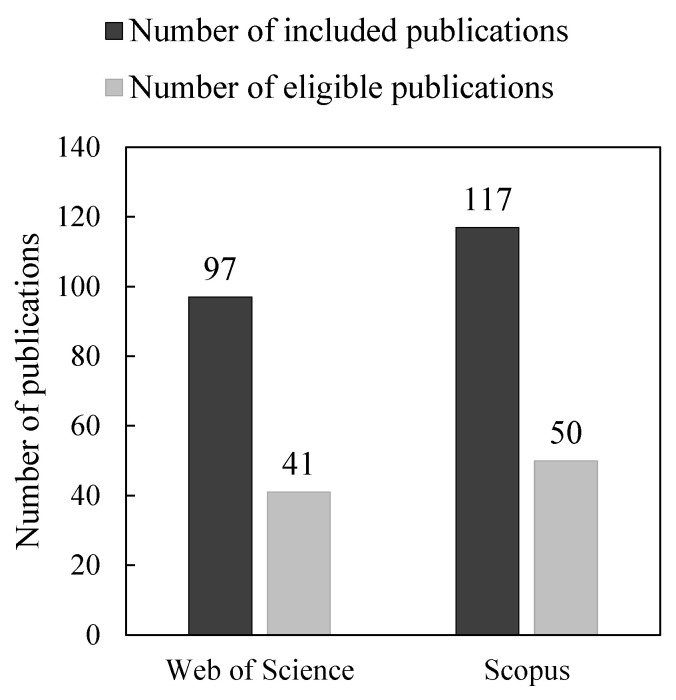
Number of publications before and after applying the exclusion criteria.

**Figure 6 bioengineering-09-00781-f006:**
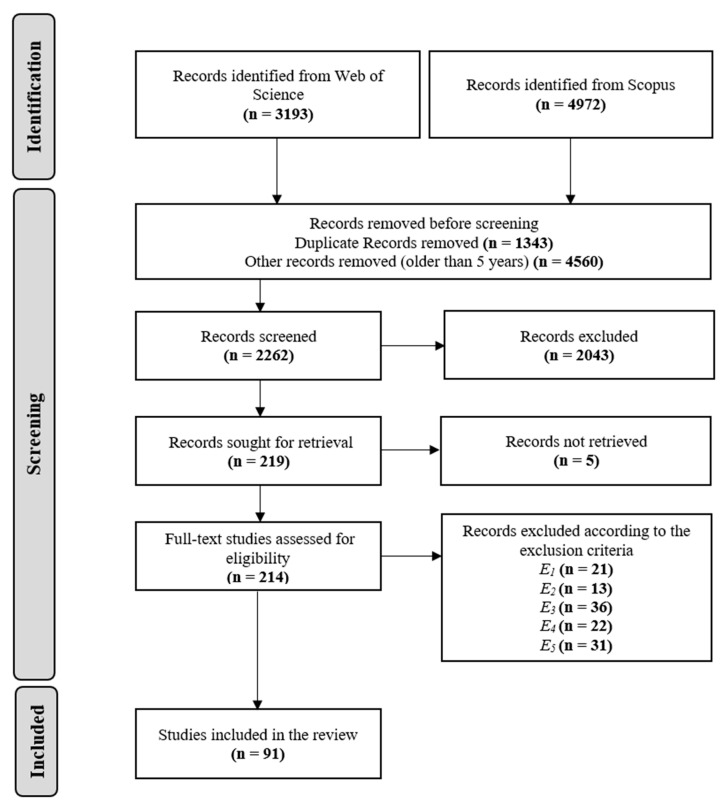
The flow diagram of articles searching and selection according to PRISMA.

**Figure 7 bioengineering-09-00781-f007:**
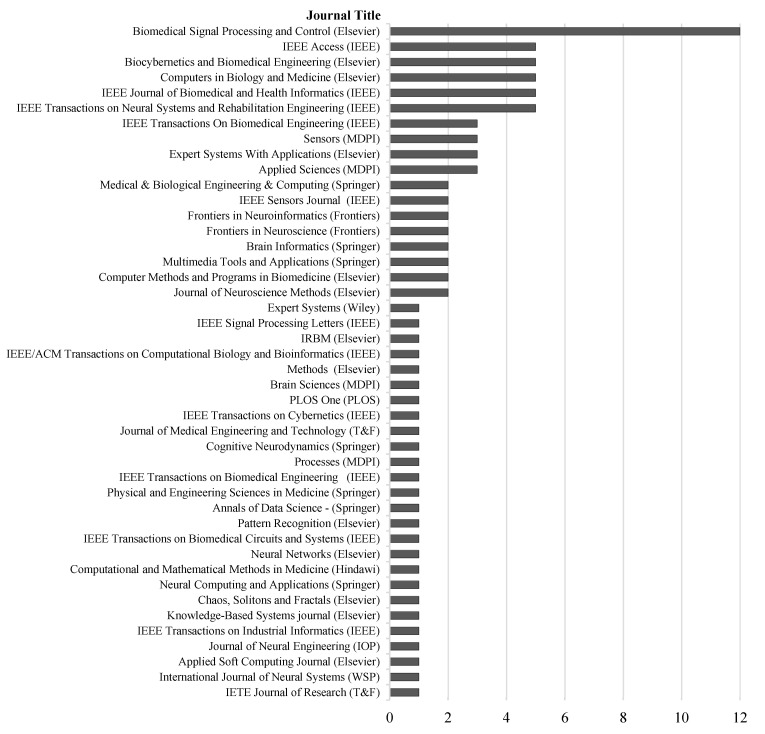
Number of publications per journal.

**Figure 8 bioengineering-09-00781-f008:**
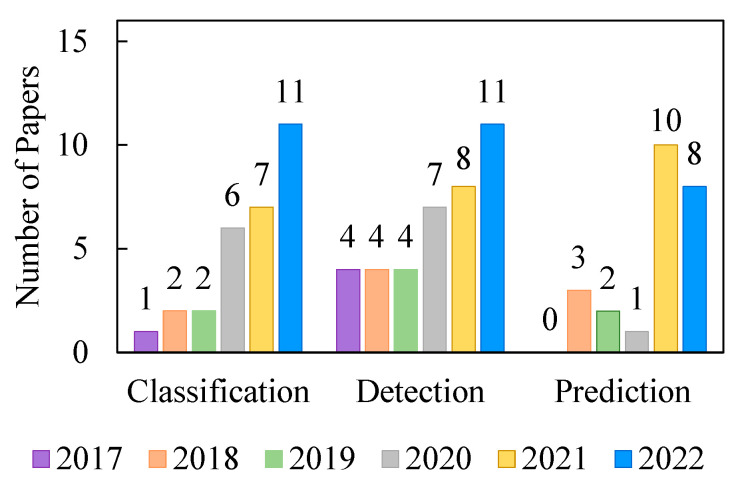
Number of publications per task.

**Figure 9 bioengineering-09-00781-f009:**
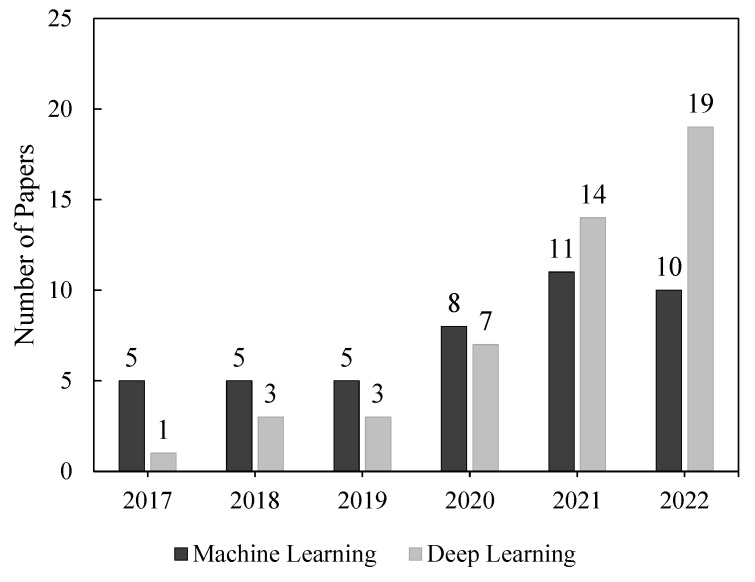
Number of publications for ML/DL methods per year.

**Table 1 bioengineering-09-00781-t001:** Overview of the top 10 cited papers for epileptic seizure recognition tasks.

Reference	Title	Publication Year	Citations
[11]	A Multivariate Approach for Patient-Specific EEG Seizure Detection Using Empirical Wavelet Transform	2017	222
[12]	Epileptic seizure detection based on EEG signals and CNN	2018	167
[13]	Neonatal Seizure Detection Using Deep Convolutional Neural Networks	2019	100
[14]	Automated seizure detection using limited-channel EEG and non-linear dimension reduction	2017	77
[15]	Epilepsy Seizure Prediction on EEG Using Common Spatial Pattern and Convolutional Neural Network	2020	68
[16]	Fuzzy distribution entropy and its application in automated seizure detection technique	2018	64
[17]	Epileptic Seizure Detection in EEG Signals Using a Unified Temporal-Spectral Squeeze-and-Excitation Network	2020	45
[18]	Generalized Stockwell transform and SVD-based epileptic seizure detection in EEG using random forest	2018	40
[19]	Deep Multi-View Feature Learning for EEG-Based Epileptic Seizure Detection	2019	36
[20]	Adaptive Multi-Parent Crossover GA for Feature Optimization in Epileptic Seizure Identification	2019	17

**Table 2 bioengineering-09-00781-t002:** Overview of the top 10 cited articles employing ML and DL for recognition of different epileptic seizures.

Group	Title	Task	Classifier	Year	Citations
Machine Learning	A Multivariate Approach for Patient-Specific EEG Seizure Detection Using Empirical Wavelet Transform [11]	Detection	RF, C4.5, FT, BayesNet NB, KNN	2017	222
	Classification of epilepsy EEG signals using DWT-based envelope analysis and neural network ensemble [21]	Classification	BPNN Ensemble	2017	123
	Automated seizure detection using limited-channel EEG and non-linear dimension reduction [14]	Detection	KNN	2017	77
	Fuzzy distribution entropy and its application in automated seizure detection technique [16]	Detection	KNN	2018	65
	Generalized Stockwell transform and SVD-based epileptic seizure detection in EEG using random forest [18]	Detection	RF	2018	40
Deep Learning	Epileptic seizure detection based on EEG signals and CNN [12]	Detection	CNN	2018	167
	Efficient Epileptic Seizure Prediction Based on Deep Learning [22]	Prediction	Deep CNN + BiLSTM	2019	130
	Neonatal Seizure Detection Using Deep Convolutional Neural Networks [13]	Detection	DCNN	2019	100
	Epilepsy Seizure Prediction on EEG Using Common Spatial Pattern and Convolutional Neural Network [15]	Prediction	CNN	2020	68
	Epileptic Seizure Detection in EEG Signals Using a Unified Temporal-Spectral Squeeze-and-Excitation Network [17]	Detection	CNN + MLP	2020	45

**Table 3 bioengineering-09-00781-t003:** Overview of the proposed solutions to the reviewed challenges.

Challenge	Solution
EEG Signal Complexity and Data Transformation	Signal engineering
High Number of EEG Channels/Channel Optimization	EEG channel reduction and attention
Generalization Ability	Adjustable training approaches, data augmentation, and various learning and training techniques
Data Imbalances	Data resampling and class balancing

**Table 4 bioengineering-09-00781-t004:** Comparison of 10 recent articles and their best performance results.

Reference	Signal Engineering	Channel Selection/Attention	Generalization Techniques	Data Balancing	Classifier	Dataset	Best Performance (%) (acc, sen, spe, pre, auc, f1) ^1^
[37]	DWT, Statistical	🗴	🗴	🗴	SVM	Bonn ^2^	97.78, 96.73, 96.79, na, na, na
[41]	Statistical	🗴	🗴	✓	DT, SVM, ANN, RF, KNN	CHB-MIT	98, 84, na, na, na, 90
						Siena-EEG	96, 84, na, na, na, 86
[45]	TQWT, Hjorth parameters	🗴	🗴	🗴	SVM	Bonn ^3^	100, 100, 100, na, na, na
[60]	STFT	🗴	🗴	✓	GTN	CHB-MIT	na, 96.01, 96.23, 95.86, na, na
[63]	DWT	✓	🗴	🗴	CNN	Bonn ^3^	100, 100, 100, na, na, na
						Bern	99.7, 99.65, 99.79, na, na, na
[70]	TQWT, Fuzzy Entropy	🗴	🗴	🗴	ANFIS	Bonn ^3^	99.83, 99.67, 99.85, 99.85, na, 99.82
						Freiburg	99.28, 99.54, 99.56, 99.29, na, 99.49
[115]	Covariance, Coherence	🗴	✓	✓	SVM	CHB-MIT	99.05, 93.56, 99.09, na, 99, na
[125]	MFCC	🗴	🗴	✓	GNN	CHB-MIT	95.38, 94.47, 94.16, na, 98.8, na
						Siena-EEG	96.05, 96.05, 96.61, na, 99.1, na
[126]	Deep features	🗴	✓	✓	CNN	CHB-MIT	96.17, 56.83, 96.97, na, na, 96.94
						TUSZ	67.68, 59.21, 75.3, na, na, 69.07
						Bonn ^3^	99.89, 99.8, 99.97, na, na, na
[132]	Deep features	🗴	🗴	🗴	GAT + BiLSTM	CHB-MIT	98.52, 97.75, 94.34, na, 96.81, 95.9
						TUSZ	98.02, 97.7, 99.06, na, 97.8, 97.86

^1^ Keywords: (acc) accuracy; (sen) sensitivity; (spe) specificity; (pre) precision; (auc) area under curve; (f1) F1-score; (na) not available. ^2^ The results are based on Healthy vs. Interictal vs. Ictal classes. ^3^ The results are based on Healthy vs. Ictal classes.

**Table 5 bioengineering-09-00781-t005:** Public epileptic seizures dataset.

	CHB-MIT	TUSZ	Bonn	Bern-Barcelona	NSC-ND	SWEC-ETHZ
Total number of seizure classes	1	8	1	1	1	1
Number of patients	23	675	23	5	10	18
Number of available channels	23–26	24–36	1	1	1	24–128
EEG type	sEEG	sEEG	sEEG/iEEG	iEEG	sEEG	iEEG
Sampling frequency	256 Hz	250 Hz	173.61 Hz	512 Hz	200 Hz	512 Hz
Total recording time	977 h	1476 h	3.2 h	41.6 h	0.2 h	2656 h
Total number of seizures	198	4029	100	3750	50	116
Detailed metadata	No	Yes	No	No	No	No

## Data Availability

Not applicable.

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
