# Peer review of "Supervised Machine Learning and Deep Learning Techniques for Epileptic Seizure Recognition Using EEG Signals—A Systematic Literature Review"

_bioengineering, 2022, doi:10.3390/bioengineering9120781_

Round 1

Reviewer 1 Report

This is a comprehensive and systematic literature review. It resembles a book chapter, and it is written in a structured way and provides very solid knowledge of Machine Learning and Deep Learning techniques. I have only a few comments.

1. Introduction needs to be revised by a neurologist.
1.1. "For this purpose, electroencephalography (EEG) was invented". Please note that EEG was introduced by Hans Berger and by other scientists as a method for recording electrical activity from healthy brains. 
1.2. There are different ictal EEG patterns in different types of epilepsy. Figure 1 shows only one example of a tonic seizure. Does this review cover other types of seizure activity, such as spike-wave discharges, epileptiform activity or interictal spikes? 
1.3. Neurologists, who are responsible for appropriate diagnostics, are skeptical about automatic recognition, even based on ML or DL. Could this issue be taken into account?

2. The meaning of "Epileptic Seizures Tasks" in the title is not clear for understanding. 

3.  Table 1 is missing (Page 7).

4. I am puzzled by the assessment of sections:
    "2. Research Methodology"
    "3. Background on Machine Learning and Deep Learning with EEG data"
    "4. Results"
    "5. Discussion"
Discussion contains conceptually important information. In my opinion, discussion is the most essential part, and I invite authors to think about rearranging their review paper.

5. This manuscript has to be corrected for redundancy and shortened.

Minor.
Page 2. "an overview of the different tasks in the field of an epileptic seizures"
Page 5. Note index 'B' in Figure 4
Page 6. "created features from constructed from the raw data"
Page 11. Figure 9, not Figure 1

Author Response

We would like to thank editors for the letter and the reviewers for their constructive comments concerning our manuscript titled “Supervised Machine Learning and Deep Learning Techniques in Epileptic Seizures Tasks using EEG Signals — A Systematic Literature Review” (ID: 1909831). We appreciate the reviewers taking the time to carefully review the manuscript and provide detailed, constructive feedback. All of these comments are beneficial and extremely helpful for revising and refining our publication, and they also have a huge guiding significance for our research. We have considered all of the comments and implemented changes that we hope will meet with your approval.

Summary of major and additional edits

We would like to inform the editors and reviewers that some addressed major changes, in addition to extra changes to the manuscript, have been amended as listed below:

  • Changes have been made to improve the flow and clarity of the manuscript, linguistically, without altering the content and are shown in red using MS Change Tracker
  • All the pages and line numbers mentioned to track the edits of the manuscript are mentioned with the MS Office “Track Changes function toggled on.
  • Some redundant content has been removed without altering the scope of the review. (Response 1.8)
  • The title is to be changed to “Supervised Machine Learning and Deep Learning Techniques for Epileptic Seizure Recognition using EEG Signals — A Systematic Literature Review” to improve the clarity of the title, according to Reviewer #1 request. (Response 1.5)
  • A few numerical values related to the counts of the articles included in the systematic review protocol have been corrected, without affecting the actual contents of the manuscript. The correct number of reviewed articles and excluded articles are 91 and 123, respectively. The values at lines 24, 360 and 362 have been updated to match the research output shown in Figure 6. (90 ⟶ 91, 124 ⟶ 123)
  • In page 12, lines 404 and 405, the counts of the articles published per mentioned journals have been corrected to match the correct statistics in Figure 7. The corrections are as follows: (22 ⟶ 26, 11 ⟶ 10 and 8 ⟶ 7)
  • Added missing journal name Computers in Biology and Medicine, Journal of Biomedical and Health Informatics. (Page 11, line 393)
  • All of the table caption numbers have been changed, as a wrong caption was present in the manuscript. (Response 1.6)
  • Figure 2 in the manuscript has been changed according to the recommendation of the neurologist who revised the “Introduction” section as requested by the reviewers. (Comments 1.0 and 3.0). This is because the existing one may be misleading in terms of highlighting seizure phases.
  • For organization purposes, some paragraphs have been moved below Figure 1. (Found on pages 3 and 4). This is because the new figure (Figure 2) is larger than the replaced one and the effect of the modifications on page spacing.

The paragraphs starting with the following sentences have been moved as it has been mentioned above:

  • “Figure 1 depicts an abstract overview of detection…..”. (Page 3, line 121)
  • “In order to comprehend the relation between these….”. (Page 4, line 130)
  • “Considering the significance of recognizing…..”. (Page 4, line 141)

  • Figure 4 resolution has been enhanced.
  • The rest of the edits in the manuscript are addressed in the following section.
  • The following new references have been added as a result to the edits made:
    • “Epilepsy.” https://www.who.int/news-room/fact-sheets/detail/epilepsy (accessed Mar. 25, 2021).
    • “Common Epilepsy Seizure Medications: Types, Uses, Effects, and More.” https://www.webmd.com/epilepsy/medications-treat-seizures (accessed Sep. 12, 2022)
    • J. Majersik et al., “A Shortage of Neurologists – We Must Act Now,” Neurology, vol. 96, no. 24, pp. 1122 LP – 1134, Jun. 2021, doi: 10.1212/WNL.0000000000012111.
    • Knowledge, Encyclopedia of Clinical Neuropsychology. 2011. doi: 10.1007/978-0-387-79948-3.
    • B. E.-S. Mohammady, “Wavelets for EEG Analysis,” Rijeka: IntechOpen, 2020, p. Ch. 5. doi: 10.5772/intechopen.94398.
    • Khan et al., “Machine Learning and Deep Learning Approaches for Brain Disease Diagnosis: Principles and Recent Advances,” IEEE Access, vol. 9, no. February, pp. 37622–37655, 2021, doi: 10.1109/ACCESS.2021.3062484.
    • W. Vinny, R. Garg, M. V. Padma Srivastava, V. Lal, and V. Y. Vishnu, “Critical appraisal of a machine learning paper: A guide for the neurologist,” Ann. Indian Acad. Neurol., vol. 24, no. 4, pp. 481–489, 2021, doi: 10.4103/aian.AIAN_1120_20.

Reviewer 2 Report

This is a well-written systematic review using PRISMA check list.

Major issue

#1. Figure 1 and 2 needs more explanation. What is the Y axis? I guess that the Y axis showed the power of Fourier transformed EEG waves.

Minor issues

#1. Since the word spasm is clearly defined term for epilepsy, muscle spasms in L3 is not appropriate. Muscle contraction is better here.

#2. Anti-epileptic drug is out of order. Currently anti-seizure medication is recommended.

#3. Please define PRISMA when first use.

Author Response

(The authors gave the same response as above.)

Reviewer 3 Report

Dear Authors,

from reading the Introduction of the paper, I note numerous inaccuracies relating to epilepsy and epileptic seizures (which I report below). I therefore stop evaluating the paper. And I invite the authors to involve a researcher expert in epilepsy in the drafting of their review. I will be happy, if the Editor deems it appropriate, to evaluate the work once it is correct from a neurological point of view.

The description of the epilepsy with which the work begins "Epilepsy is a neurological disorder that can cause jerking of various parts of the body and, in some cases, uncontrollable muscle spasms across the entire body" is reductive compared to the numerous and polymorphic manifestations of epileptics seizures, and does not correspond to the definitions currently in use neither of epileptic seizures nor of epilepsy.

"Seizure detection" as binary decision is an optimistic view. Some discharges, especially in certain contexts (eg the “ictal-interictal continuum” pattern, certainly pathological, indicates suspicion of critical activity, not certainty).

The statement "The interictal phase describes the normal state of the brain between two consecutive seizures, where no epileptiform EEG activity is present" is incorrect, or at least strongly equivocal. The interictal phase is characterized by the absence of seizures, but by the presence - in many but not all patients - of epileptiform graphoelements, that is, elements that are significant for the diagnosis but are subclinical.

Other significant inaccuracies are contained in the following paragraph.

In Figures 1 and 2, the unit of measurement of the ordinates and abscissas is not indicated.

Sincerely.

Author Response

Please see the point to point reply to your comments  in the attached file.

We would like to thank editors for the letter and the reviewers for their constructive comments concerning our manuscript titled “Supervised Machine Learning and Deep Learning Techniques in Epileptic Seizures Tasks using EEG Signals — A Systematic Literature Review” (ID: 1909831). We appreciate the reviewers taking the time to carefully review the manuscript and provide detailed, constructive feedback. All of these comments are beneficial and extremely helpful for revising and refining our publication, and they also have a huge guiding significance for our research. We have considered all of the comments and implemented changes that we hope will meet with your approval.

Summary of major and additional edits

We would like to inform the editors and reviewers that some addressed major changes, in addition to extra changes to the manuscript, have been amended as listed below:

  • Changes have been made to improve the flow and clarity of the manuscript, linguistically, without altering the content and are shown in yellow highlights using MS Change Tracker
  • All the pages and line numbers mentioned to track the edits of the manuscript are mentioned with the MS Office “Track Changes function toggled on.
  • Some redundant content has been removed without altering the scope of the review. (Response 1.8)
  • The title is to be changed to “Supervised Machine Learning and Deep Learning Techniques for Epileptic Seizure Recognition using EEG Signals — A Systematic Literature Review” to improve the clarity of the title, according to Reviewer #1 request. (Response 1.5)
  • A few numerical values related to the counts of the articles included in the systematic review protocol have been corrected, without affecting the actual contents of the manuscript. The correct number of reviewed articles and excluded articles are 91 and 123, respectively. The values at lines 24, 360 and 362 have been updated to match the research output shown in Figure 6. (90 ⟶ 91, 124 ⟶ 123)
  • In page 12, lines 404 and 405, the counts of the articles published per mentioned journals have been corrected to match the correct statistics in Figure 7. The corrections are as follows: (22 ⟶ 26, 11 ⟶ 10 and 8 ⟶ 7)
  • Added missing journal name Computers in Biology and Medicine, Journal of Biomedical and Health Informatics. (Page 11, line 393)
  • All of the table caption numbers have been changed, as a wrong caption was present in the manuscript. (Response 1.6)
  • Figure 2 in the manuscript has been changed according to the recommendation of the neurologist who revised the “Introduction” section as requested by the reviewers. (Comments 1.0 and 3.0). This is because the existing one may be misleading in terms of highlighting seizure phases.
  • For organization purposes, some paragraphs have been moved below Figure 1. (Found on pages 3 and 4). This is because the new figure (Figure 2) is larger than the replaced one and the effect of the modifications on page spacing.

The paragraphs starting with the following sentences have been moved as it has been mentioned above:

  • “Figure 1 depicts an abstract overview of detection…..”. (Page 3, line 121)
  • “In order to comprehend the relation between these….”. (Page 4, line 130)
  • “Considering the significance of recognizing…..”. (Page 4, line 141)

  • Figure 4 resolution has been enhanced.
  • The rest of the edits in the manuscript are addressed in the following section.
  • The following new references have been added as a result to the edits made:
    • “Epilepsy.” https://www.who.int/news-room/fact-sheets/detail/epilepsy (accessed Mar. 25, 2021).
    • “Common Epilepsy Seizure Medications: Types, Uses, Effects, and More.” https://www.webmd.com/epilepsy/medications-treat-seizures (accessed Sep. 12, 2022)
    • J. Majersik et al., “A Shortage of Neurologists – We Must Act Now,” Neurology, vol. 96, no. 24, pp. 1122 LP – 1134, Jun. 2021, doi: 10.1212/WNL.0000000000012111.
    • Knowledge, Encyclopedia of Clinical Neuropsychology. 2011. doi: 10.1007/978-0-387-79948-3.
    • B. E.-S. Mohammady, “Wavelets for EEG Analysis,” Rijeka: IntechOpen, 2020, p. Ch. 5. doi: 10.5772/intechopen.94398.
    • Khan et al., “Machine Learning and Deep Learning Approaches for Brain Disease Diagnosis: Principles and Recent Advances,” IEEE Access, vol. 9, no. February, pp. 37622–37655, 2021, doi: 10.1109/ACCESS.2021.3062484.
    • W. Vinny, R. Garg, M. V. Padma Srivastava, V. Lal, and V. Y. Vishnu, “Critical appraisal of a machine learning paper: A guide for the neurologist,” Ann. Indian Acad. Neurol., vol. 24, no. 4, pp. 481–489, 2021, doi: 10.4103/aian.AIAN_1120_20.   

Please see the point to point reply to your comments  in the attached file.

Round 2

Reviewer 3 Report

There are some trivial errors: a) two “Figure 1” (page 2 and page 11); b) in the text (line 150 and line 194) the quotation of figures 5 and 4 seems to be reversed.

To move on to more substantive issues, there is the fact that the review says very little about the performance of the various methods under review in correctly identifying critical patterns; and in the concordance of the results with those obtained with the visual survey of the tracings (no other diagnostic gold standard is possible). A clarification on the state of the art in this regard is necessary.

Author Response

Reviewers’ comments and responses

Reviewer #3

Comment 3.1: There are some trivial errors: a) two “Figure 1” (page 2 and page 11); b) in the text (line 150 and line 194) the quotation of figures 5 and 4 seems to be reversed.

To move on to more substantive issues, there is the fact that the review says very little about the performance of the various methods under review in correctly identifying critical patterns; and in the concordance of the results with those obtained with the visual survey of the tracings (no other diagnostic gold standard is possible). A clarification on the state of the art in this regard is necessary.

Response 3.1: We thank the reviewer for the suggestions and comment. Figure 9 numbering has been fixed in page 11, and the quotations for both figures 4 and 5 have been corrected.

We would like to clarify that the systematic review scope is to focus on the challenges and methods used with supervised machine and deep learning techniques with different recognition tasks (detection, prediction, and classification). The review incorporates the most recent strategies and attempts offered by other researchers on how to overcome the difficulties encountered while working with clinical data linked to seizures using EEG data. It is crucial to highlight that the review mainly focuses on supervised learning approaches, which require the existence of expert annotations on the EEG signals in the data utilized during the learning process. This type of information and data (datasets) are the ones used by the articles under review, and a description of the frequently used datasets is presented in Section 6 of the review. These datasets are annotated by experts / neurologists, whose expertise is regarded the gold standard for comparing the performance results of the evaluated articles against. Since the data used is already annotated by the previously mentioned personnel, none of the reviewed articles considered a validation process that includes measuring the performance of a proposed method against the visual inspection done by experts. Performance evaluations that require a separate validation process from experts through visual inspection are better suited with unsupervised learning techniques, where no validation data is available to validate the proposed method’s performance against. In this case, experts’ opinion on the results is deemed necessary as it is considered the ground truth. Unsupervised learning approaches of this type are out of the scope of this review.

As per the reviewer’s remark, sub-section “5.3” named “Performance comparison” has been added. This sub-section covers the performance results of the 10 recent articles, published in 2022, which are reviewed in the paper according to different evaluation metrics across different datasets (pages 26 and 27). Additionally, this sub-section includes a summary of the selected articles’ contributions addressing the surveyed challenges. For the sake of a constructive comparison, the articles selected reported their performance results using at least 3 out of 6 evaluation metrics. These metrics are accuracy, sensitivity, specificity, precision, area under curve (AUC), and F1-score.

Additional Information: The following excerpts are copied verbatim from TUH Corpus documentation material regarding the annotation procedure of seizures using EEG data [1]:

“Our approach to annotation of EEG signals has evolved since our initial work in 2012. We have worked closely with a team of neurologists at Temple Hospital to understand their workflow and their clinical needs. We have refined our annotation process to better characterize their clinical needs and the needs of our machine learning technology. These standards were developed to consistently distinguish seizure, slowing, and artifact events with a high degree of accuracy and precision”

“The development of a team of undergraduates that can accurately annotate seizures has been an interesting journey (Shah et al., 2020). Annotation is carried out by a team dedicated solely to EEG interpretation. These annotators are Temple University students who have undergone several months of rigorous training in order reach annotation skills on par with our standards. They must be able to recognize and classify seizure and artifact events with a high degree of interrater reliability so as to maintain the integrity of the TUEG. These students usually have a STEM background, and often are pursuing degrees in neuroscience or bioengineering.

During the first round of annotation, a single annotator will check each file for seizure events. If an annotator is unable to make a definitive judgment on an event, the annotator will mark this file for a review from a more senior member of the group. If the senior member is unable to discern whether the event is in fact seizure or not, that senior member will mark this file to be reviewed during a weekly meeting. At this meeting, events are annotated on a consensus basis.

Following the completion of the first round of annotation for every file in the set, a round of reviews will begin. For a new data set, each file will be reviewed by two annotators individually. This redundancy is in place to reduce the number of missed events, so that all precious seizure data is harvested, and all annotations are accurate, orderly, and meet our standards.

For a data set that has already undergone the first set of annotation and review, sometimes a revision is in order. Revisions may be carried out on the entirety of the set, as is sometimes the case for an old set which needs to be brought up to current standards, or for a subset of the data set. This may be a review of seizure files only, which acts to ensure the highest accuracy of our seizure data and screen out any false alarms, or a review of files marked as seizure by the machine learning system, to find missed seizures and reduce the rate of false negatives. These reviews are done to ensure the best possible accuracy for the data. The corpus has undergone numerous revisions by the annotation team, each time refining the data and implementing more precise and exact standards to enhance the clarity and accuracy of the data.

Revision is done by either a two or three annotator per file system. The first annotator reviews the previous annotations for accuracy. If this annotator decides no change is necessary, the file is then marked as such. Another annotator will check this file. If the second annotator agrees, the file is then marked as complete with no necessary changes. If the second annotator does not agree, it will be checked by a third annotator as a tiebreaker. If the first annotator decides a change is necessary, they will make the appropriate changes. These changes will be reviewed by the second and third annotators. If they are affirmed, the changes will be kept. If not, the three annotators will review the file and come to a consensus.”

References

[1] “Index of /publications/reports/2020/tuh_eeg/annotations.” https://isip.piconepress.com/publications/reports/2020/tuh_eeg/annotations/ (accessed Sep. 26, 2022).

Please see the details in the attached file.

Round 3

Reviewer 3 Report

Dear Editor, Dear Authors,

Improvements were made to the manuscript both in neurological aspects and in the methods of evaluating the performance of the algorithms used for reading the EEG traces.

I appreciated the submission of the Annotation guidelines of The Temple University Hospital EEG Corpus. However, I note that they do not take into account the nomenclature currently in use (according to the criteria of the International Ligue against Epilepsy, ILAE) and that in Table 1 (The labels) some of the labels refer to entities that cannot be identified only on the basis of EEG, but which require a concomitant clinical evaluation (example: Simple partial seizure, Complex partial seizure; terms - moreover - no longer recommended and replaced by other terms). These limitations should be mentioned in the manuscript. This constitutes a further reason to recommend closer collaboration between bioengineers and neurologists.

Sincerely.

Author Response

Acknowledgement

Dear Editors and Reviewers,

We would like to thank editors for the letter and the reviewers for their constructive comments concerning our manuscript titled “Supervised Machine Learning and Deep Learning Techniques in Epileptic Seizures Tasks using EEG Signals — A Systematic Literature Review” (ID: 1909831). We appreciate the reviewers taking the time to carefully review the manuscript and provide detailed, constructive feedback to enhance the quality of this paper.

Summary of Changes

Reviewers’ comments and responses

Reviewer #3

Comment 3.1: Dear Editor, Dear Authors,

Improvements were made to the manuscript both in neurological aspects and in the methods of evaluating the performance of the algorithms used for reading the EEG traces.

I appreciated the submission of the Annotation guidelines of The Temple University Hospital EEG Corpus. However, I note that they do not take into account the nomenclature currently in use (according to the criteria of the International Ligue against Epilepsy, ILAE) and that in Table 1 (The labels) some of the labels refer to entities that cannot be identified only on the basis of EEG, but which require a concomitant clinical evaluation (example: Simple partial seizure, Complex partial seizure; terms - moreover - no longer recommended and replaced by other terms). These limitations should be mentioned in the manuscript. This constitutes a further reason to recommend closer collaboration between bioengineers and neurologists.

Sincerely.

Response 3.1:

Dear Editor, Dear Reviewer,

We would like to thank you for all the time and effort you put into providing improvements and recommendations to enhance the quality of this paper. We appreciate the comprehensive review of all elements of the paper, both from neurological and bioinformatic aspects.

Regarding the mentioned limitations by the reviewer commenting on the annotation guidelines of The Temple University Hospital EEG Corpus, we found out that the annotation guidelines has explicitly mentioned that simple partial and complex partial seizures cannot solely identified based on EEG signals, according the following text [1]:

“Complex partial seizures are focal seizures in which the patient does not maintain awareness. Again, this can only be indicated with clinical data captured in the EEG report.”

This conforms with the reviewer’s comment.

Regarding the part concerning the outdated terms that are not following the recent nomenclature, we have added the following text (in page 29, line 1175) and reference:

  • Despite the TUSZ dataset comprises several types of epileptic seizures, some seizure terms do not conform to the most recent nomenclature as determined by the International League Against Epilepsy (ILAE). Simple partial and complex partial seizures are examples of these seizures, which are currently designated as focal aware and focal impaired awareness seizures, respectively [147]. Moreover, these types of seizures require further clinical evaluation as they cannot be solely identified through EEG analysis. These limitations emphasize the importance of increased collaboration between bioengineers and neurologists, as well as discovering new approaches, using ML and DL, that incorporate the use of clinical reports along with EEG analysis to gain deeper knowledge about similar types of seizures.

References added:

  • S. Fisher et al., “Instruction manual for the ILAE 2017 operational classification of seizure types,” Epilepsia, vol. 58, no. 4, pp. 531–542, 2017, doi: 10.1111/epi.13671.

References

[1]       “Index of /publications/reports/2020/tuh_eeg/annotations.” https://isip.piconepress.com/publications/reports/2020/tuh_eeg/annotations/ (accessed Sep. 26, 2022).
